# Structures of the holo CRISPR RNA-guided transposon integration complex

Jung-Un Park[1,2], Amy Wei-Lun Tsai[1,2], Alexandrea N. Rizo[1,2], Vinh H. Truong[1], Tristan X. Wellner[1], Richard D. Schargel[1] & Elizabeth H. Kellogg[1✉]

CRISPR-associated transposons (CAST) are programmable mobile genetic elements that insert large DNA cargos using an RNA-guided mechanism[1–3]. CAST elements contain multiple conserved proteins: a CRISPR effector (Cas12k or Cascade), a AAA+ regulator (TnsC), a transposase (TnsA–TnsB) and a target-site-associated factor (TniQ). These components are thought to cooperatively integrate DNA via formation of a multisubunit transposition integration complex (transpososome). Here we reconstituted the approximately 1 MDa type V-K CAST transpososome from *Scytonema hofmannii* (*Sh*CAST) and determined its structure using single-particle cryo-electon microscopy. The architecture of this transpososome reveals modular association between the components. Cas12k forms a complex with ribosomal subunit S15 and TniQ, stabilizing formation of a full R-loop. TnsC has dedicated interaction interfaces with TniQ and TnsB. Of note, we observe TnsC–TnsB interactions at the C-terminal face of TnsC, which contribute to the stimulation of ATPase activity. Although the TnsC oligomeric assembly deviates slightly from the helical configuration found in isolation, the TnsC-bound target DNA conformation differs markedly in the transpososome. As a consequence, TnsC makes new protein–DNA interactions throughout the transpososome that are important for transposition activity. Finally, we identify two distinct transpososome populations that differ in their DNA contacts near TniQ. This suggests that associations with the CRISPR effector can be flexible. This *Sh*CAST transpososome structure enhances our understanding of CAST transposition systems and suggests ways to improve CAST transposition for precision genome-editing applications.

CAST are Tn7-like transposons[1] that programmably integrate large DNA cargos at genomic locations dictated by a guide RNA sequence[2,3]. Multiple independent acquisition events[1,4] created the distinct CAST subfamilies characterized to date: I-F3[3,5], I-B[6] and V-K CAST[2]. One of the most remarkable aspects of CAST function is the precision of the programmable insertions: DNA cargo is inserted in a single orientation, with defined spacing from the protospacer adjacent motif (PAM) and within a narrow window (5–10 bp). These features appear to be generally conserved across CAST families[3,5–7], and are probably a consequence of the transpososome architecture (that is, the nucleoprotein integration complex containing all CAST components).

Across all CAST elements[4], four or more proteins make up the CAST transposition machinery: the CRISPR effector (Cas12k or Cascade), TniQ, TnsC and the transposase (TnsB or TnsA–TnsB). In all Tn7-like transposition systems, a specialized CRISPR effector or DNA-binding protein (TnsD in the case of the prototypic Tn7)[8] is hypothesized to recruit core transposition proteins to the target site[9]. In CAST elements, the TniQ protein (which is related to TnsD) associates with the target site via interactions with the CRISPR effector[10]. The AAA+ regulator TnsC serves as a molecular matchmaker by interacting with both

target-site-binding proteins and the transposase[11,12]. Very little structural information exists to explain how TnsC bridges the components in the hypothesized transpososome assembly. Furthermore, how the core transposition machinery evolved to adopt different targeting strategies[6,13,14] remains an open question.

Owing to its simplicity and robust in vitro activity[2], ShCAST is an attractive system for understanding the general mechanisms used by CAST elements. Although the structure of each *Sh*CAST component has been individually characterized[15–18], the spatial associations across the entire transposition recruitment process remain unknown. We reconstituted the *Sh*CAST transpososome by mimicking the natural configuration found in programmed integration (Fig. 1a). Using single-particle cryo-electron microscopy (cryo-EM), we obtained a high-resolution structure (3.5 Å resolution) that was sufficient to accurately resolve the interactions between all components within the 961 kDa nucleoprotein complex.

We found that the transpososome architecture promotes modular association across CAST components, and TnsC has dedicated faces of interaction with TniQ and TnsB, consistent with models of Tn7 transposition[12,19]. Cas12k, TniQ and S15 stabilize R-loop formation, but TniQ and

[1]Department of Molecular Biology and Genetics, Cornell University, Ithaca, NY, USA. [2]These authors contributed equally: Jung-Un Park, Amy Wei-Lun Tsai, Alexandrea N. Rizo.
✉e-mail: lizkellogg@gmail.com

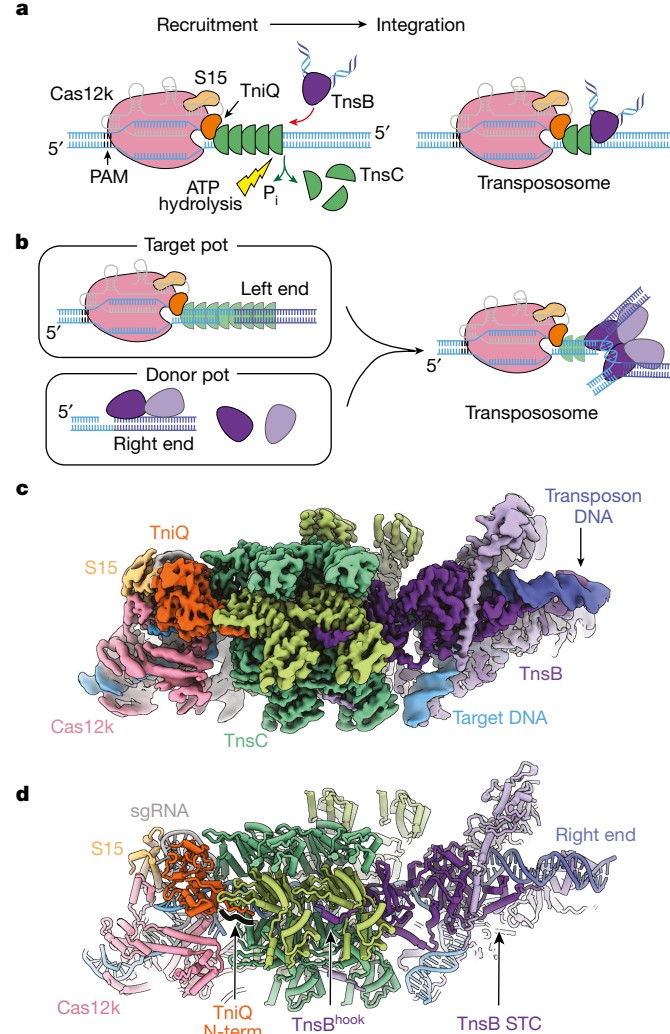

**Fig. 1 | Cryo-EM structure of *Sh*CAST transpososome. a**, Mechanistic model of *Sh*CAST recruitment (left) and integration (right). TnsC (green semicircle) associates with target-site proteins: Cas12k (pink), TniQ (orange) and S15 (tan). The PAM (black) defines the beginning of the protospacer, where Cas12k and single guide RNA (sgRNA) (grey) bind. TnsB (purple) is recruited to the target site through TnsC (red arrow). ATP hydrolysis (yellow lightning bolt) is stimulated by TnsB, resulting in the release of phosphate (P$_i$) and disassembly of TnsC (green arrows). Upon integration, a nucleoprotein complex containing all CAST components forms at the target site (right). **b**, Schematic of transpososome sample preparation. Left, the target pot and the donor pot were prepared independently to mimic the process of the RNA-guided transposition. DNA substrates for both the target pot and donor pot have target DNA (light blue) and transposon-end (dark blue) regions that are connected by 5 bp of single-stranded DNA, designed to form a strand-transfer complex (STC) (Methods). The target pot and donor pot include target-site-associated proteins (Cas12k, S15, TniQ and TnsC) and TnsB, respectively, in addition to the corresponding DNA substrate. The transpososome (right) is reconstituted by combining the target and donor pots (Methods). **c**, The 3.5 Å-resolution cryo-EM reconstruction of the *Sh*CAST transpososome, filtered according to local resolution. Each component in the complex is coloured as in **a**. Different shades (light or dark green and light or dark purple) indicate different subunits of the protein of the same colour. Target DNA is shown in light blue and transposon DNA is in dark blue. **d**, The atomic model of the *Sh*CAST transpososome. TniQ N terminus comprises residues 1–10. Right and left refer to the ends of the transposon DNA.

TnsC form the primary connection between the CRISPR effector and transposase. We also observed TnsB–TnsC interactions that are important for TnsB-promoted ATP hydrolysis. Notably, in the context of the

transpososome, TnsC protomers make functionally important interactions with the target DNA that are distinct from interactions found in helical TnsC[18]. Finally, we found that the transpososomes assemble with a slightly different number of TnsC protomers (±1 protomer), hinting at the molecular basis of the intrinsic variability of *Sh*CAST insertions. The structural insight gained from this transpososome structure contributes to the understanding of CAST transposition and serves to identify avenues for protein engineering.

## DNA design and transpososome reconstitution

*Sh*CAST insertions occur within a 61- to 66-base pair (bp) window downstream from the PAM in a single orientation[2,18]. Because transposition is driven primarily via protein–DNA interactions, the transpososome is the most stable structure in the transposition pathway. This principle has been applied successfully in the past to stabilize STC structures of other integrases and transposases, such as PFV[20,21]. Thus, we designed a strand-transfer substrate (Extended Data Fig. 1a) that contains binding sites for all *Sh*CAST components[2]. Our designed DNA substrate contains transposon DNA up to the first two internal TnsB binding sites from the right and left ends (Extended Data Fig. 1b), identical to previous studies[17,22]. The first 30 base pairs have identical TnsB binding sites on either end[17], in contrast to the subsequent internal TnsB binding sites that are irregularly spaced in the right transposon end compared with the left transposon end. Therefore, the transposon sequences that we included in our designed substrate most probably do not contribute to the remarkable ability of *Sh*CAST to discriminate the insertion orientation[2]. The 'target pot' from the transpososome reconstitution procedure (Fig. 1b) is similar to the procedure used to reconstitute a *Sh*CAST subcomplex referred to as the 'recruitment complex'[23], which contains all the components except TnsB. We also based our reconstitution procedure on established methods for monitoring transposition in vitro[18,24] and reasoned that this procedure would mimic the recruitment and integration process (Fig. 1b). Incubation of target DNA with target-site-binding proteins (Cas12k, TniQ, S15 and TnsC) and ATP resulted in a heterogeneous assembly, comprising different TnsC filament lengths (as assessed by negative-stain electron microscopy; Extended Data Fig. 1c). Subsequent incubation with transposon-DNA-bound TnsB, followed by enrichment for DNA-bound complexes resulted in the successful reconstitution of complete transpososome particles (Fig. 1b and Extended Data Fig. 1d), which we analysed by high-resolution cryo-EM imaging and image analysis (Methods and Extended Data Fig. 2).

## Overall transpososome architecture

A 3.5 Å cryo-EM reconstruction (Extended Data Table 1) of the reconstituted *Sh*CAST transpososome reveals an assembly composed of one Cas12k subunit, one TniQ subunit, one ribosomal S15 subunit, four subunits of TnsB, and two full turns of TnsC with six subunits per turn (Fig. 1c). The defined oligomeric assembly of TnsC is important, because the *Sh*CAST recruitment complex (containing all components except TnsB) consists of heterogeneous assemblies of TnsC, distinguished by the direction of TnsC filaments bound to DNA[23] and the number of turns of TnsC. In contrast to the heterogeneity of the recruitment complex (captured in the target pot reconstitution, Extended Data Fig. 1c), transpososome particles reveal uniform TnsC directionality (Fig. 1c,d), in which the N-terminal face of TnsC interacts with TniQ at the target site and the C-terminal face of TnsC interacts with TnsB (Extended Data Fig. 3). In addition, TnsC filaments in transpososome particles are of uniform stoichiometry, consisting of two turns of TnsC. TnsC is disassembled by TnsB during recruitment[18] (Fig. 1a). Therefore, the homogeneity of the TnsC oligomeric assembly that we observe in our transpososome particles suggests that (1) TnsC protomers lacking productive interactions with target-site proteins are presumably disassembled by TnsB, and (2) TnsB disassembles TnsC filaments until a

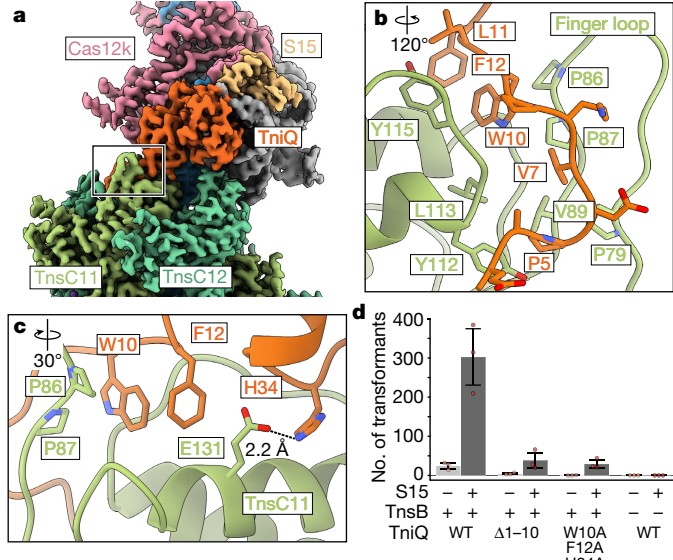

**Fig. 2 | TniQ–TnsC interactions are crucial for *Sh*CAST transposition. a**, The composite cryo-EM map was created using two separate reconstructions from the local refinements (Methods). The composite map is coloured as in Fig. 1. TnsC protomers are labelled according to their position in the transpososome (TnsC1 is the closest TnsC protomer to TnsB). The outline indicates the TniQ–TnsC interface. **b,c**, The atomic model of interactions at the TniQ–TnsC interface rotated 120° around the vertical axis (**b**) and 30° around the horizontal (**c**) with respect to the outlined region in **a**. **b**, The N-terminal tail of TniQ is shown—as sticks—interacting with a hydrophobic cleft created by the finger loop of TnsC. **c**, Hydrogen-bonding interactions are shown as dashed black lines; distance between the donor and acceptor atoms is indicated. **d**, TniQ mutations eliminate in vitro transposition activity, consistent with the interactions observed in our transpososome structure. WT, wild type. In vitro transposition activity was monitored by transforming the reaction product into competent cells, and counting the number of transformants after plating on an antibiotic-containing plate (Methods). Data are mean ± s.d. (*n* = 3 biological triplicates). Raw data points are shown in red.

certain point, reflected in the uniform stoichiometry of TnsC oligomers observed in our transpososome reconstitution. Therefore, the interactions between TnsC and target-site-associated proteins (Cas12k, TniQ and S15) are stabilized against further disassembly by TnsB, probably owing to interactions at the target site (Fig. 1a, right).

## Transposition requires TniQ–TnsC contacts

Owing to slight flexibility in the DNA substrate, the distal ends of the transpososome exhibit lower local resolution in our structure (5–7 Å; Extended Data Fig. 2e). As expected, local refinement focused on the Cas12k-proximal region improved the quality of the reconstruction (Supplementary Fig. 1a,b). Consistent with the functional importance of TniQ in *Sh*CAST transposition[16], TniQ is located at the PAM-distal end of the R-loop close to Cas12k and S15 (Extended Data Fig. 4a). TniQ primarily interacts with TnsC and RNA, consistent with the productive recruitment complex[23]. S15 is nestled between the REC2 domain of Cas12k and the PAM-distal sgRNA–DNA heteroduplex (Extended Data Fig. 4d). The rooftop loop of the sgRNA is flanked on either side by S15 and TniQ, respectively. TniQ bridges the two TnsC protomers closest to Cas12k (Fig. 2a and Supplementary Fig. 2), however the TniQ–TnsC12 interface is much smaller than the TniQ–TnsC11 interface (325 Å$^2$ versus 915 Å$^2$; Methods). More specifically, the N-terminal tail of TniQ forms structured interactions with the finger loop of TnsC11 and is generally hydrophobic in nature (Fig. 2b). From the structure, we identified hydrophobic residues W10 and F12, which are buried into a hydrophobic cleft

in TnsC11 formed by the finger loop (Fig. 2b), as important residues. H34 forms hydrogen-bonding interactions with the acidic residue E131 on the N-terminal face of TnsC11 (Fig. 2c). Truncation of the first ten residues or mutation of the hydrophobic residues W10, F12 and H34 results in a near-complete loss of transposition activity (Fig. 2d), consistent with the essential role suggested by our structural observations. Consistent with the importance of S15 for formation of the productive recruitment complex[23], we find that S15 generally improves transposition activity (Fig. 2d). However, the addition of S15 does not change our conclusions regarding the importance of the TniQ residues (Fig. 2d). Cas12k–TnsC interactions are completely absent (Extended Data Fig. 4c), and 3D variability analysis suggests that Cas12k enables flexible linkage to the other *Sh*CAST components (Supplementary Video 1). Together, these observations collectively suggest that the TniQ–TnsC interactions that we observe are the primary connections required to direct insertions to a guide RNA-directed target site. This supports the idea that engineering new associations with target-site recognition modules (that is, novel non-CAST CRISPR effectors) can be achieved by focusing on TniQ.

## TnsB forms structured interactions with TnsC

We next focused on the interactions between TnsB and TnsC in the transpososome structure (Fig. 3a). TnsB belongs to the DDE/D transposase family, and bears significant similarities to MuA from bacteriophage Mu[17,22,25,26]. TnsB contains multiple functionally distinct domains dedicated to DNA-binding, catalytic activity and interactions with TnsC[17] (Supplementary Fig. 3). The previously determined tetrameric TnsB STC structure[17] (PDB: 7SVW) docks well into the transpososome cryo-EM map, requiring only minimal modifications throughout (1.54 Å Cα root mean squared deviation (r.m.s.d.), Supplementary Fig. 4). Full-length TnsB has a 52-residue-long flexible linker (residues 518–569), that connects the STC to the C-terminal 'hook' (TnsB$^{hook}$, residues 570–584) (Supplementary Fig. 3). TnsB$^{hook}$ serves as the point of contact with the TnsC filament body and is critically important for CAST transposition[17]. We identified well-defined density corresponding to the TnsB$^{hook}$ at 4 of the 6 TnsC protomers closest to TnsB (TnsC protomers 2–5) (Fig. 3b and Extended Data Fig. 5). After docking in TnsB STC structure[17], we found additional unassigned density corresponding to TnsB residues 475–542. These residues, which were previously missing in STC and predicted to be disordered by DISOPRED3[27] (Supplementary Fig. 3), form a helix-turn-helix motif (Fig. 3c). Therefore, although TnsB–TnsC associations may be flexible in principle, we see very well-defined TnsB–TnsC interactions in the transpososome, with all four TnsB protomers engaged at precise locations on the TnsB-proximal TnsC hexamer.

## TnsB domain IIβ stimulates ATPase activity

In the *Sh*CAST system, TnsB is recruited to the target site via TnsC[17] (Fig. 1a, left). This is intrinsically tied to ATP hydrolysis, because substitution with a non-hydrolyzable ATP analogue (AMP-PNP) inhibits TnsB-mediated filament disassembly and results in non-targeted transposition[18]. Therefore, a crucial role of TnsB is also to stimulate the ATPase activity of TnsC in the process of target-site selection. In prototypic Tn7, C-terminal fragments are sufficient to stimulate TnsC ATPase activity[28]. In *Sh*CAST, two TnsB–TnsC interactions have been biochemically demonstrated to be required for this process, one of which corresponds to TnsB$^{hook}$ and the other is located elsewhere in full-length TnsB[17]. Notably, within the transpososome structure, we observe a second TnsB–TnsC interaction at the C-terminal face of TnsC (Fig. 3c,d). In particular, TnsB residues 410–542 (including part of a flexible linker) are positioned to bridge two TnsC protomers (Fig. 3c). Notably, domain IIβ (residues 410–474) localizes close to the ATP-binding pocket of TnsC, abutting against helix α4 in TnsC (Fig. 3d). This helix contains functionally important, conserved residues Q185 and R189 (Fig. 3d), which recognize the nucleotide-bound state[18].

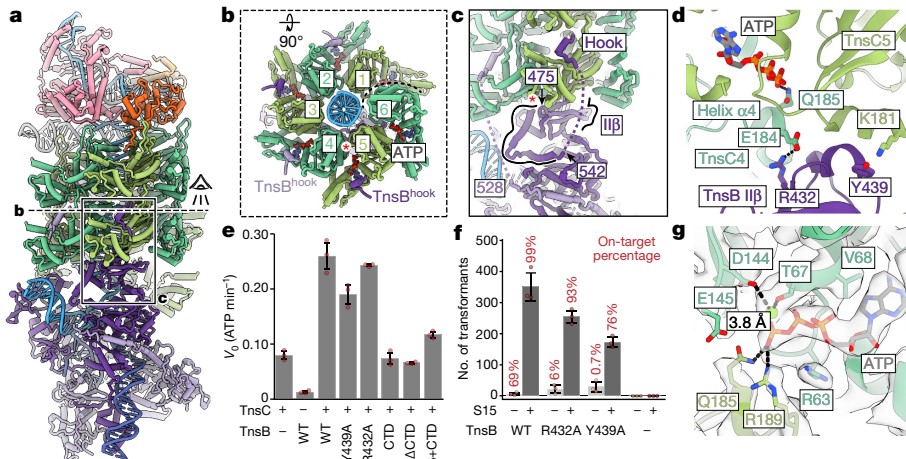

**Fig. 3 | TnsB–TnsC interactions are well defined and contribute to stimulating the ATPase activity of TnsC. a**, An overview of TnsB–TnsC interactions in the transpososome. **b**, The N-terminal face of the TnsC hexamer closest to TnsB, as indicated in **a**. TnsC protomers are labelled according to their position (1 indicates the TnsC protomer closest to TnsB). The dashed line indicates where the hexamer ends. TnsB^hook peptides bind at TnsC protomers 2–5 and are coloured according to nearest TnsB protomer. The red asterisk indicates where domain IIβ (residues 410–474) associates across 2 TnsC protomers on the C-terminal face of TnsC. **c**, Side view of TnsB–TnsC interactions, as indicated in **a**. Residues 475–542 correspond to the additional structured TnsB domain (highlighted with black lines) that is observed to interact between the two TnsC protomers shown in **b**. The red asterisk indicates the beginning of helix α4. Dotted lines indicate the flexible linker (not observed in our structure).

**d**, TnsB–TnsC interactions near the ATP-binding pocket. Interacting residues are shown and dashed lines represent hydrogen bonds. **e**, Alterations to TnsB domain IIβ decrease ATP hydrolysis activity. The hydrolysis rate ($v_o$) is shown for each variant. Data are mean ± s.d. ($n = 3$ biological triplicates). **f**, In vitro transposition assay testing TnsB mutants in the presence (dark grey) or absence (light grey) of S15. The number of transformants is plotted for each condition tested as a proxy for the transposition activity. Data are mean ± s.d. ($n = 3$ biological triplicates). Raw data points are shown in red. The on-target percentage of transposition was estimated from Illumina sequencing and indicated on the corresponding bar plot. **g**, The ATP-binding pocket of TnsC with cryo-EM density (grey with transparent surface), magnesium ion (green sphere) and hydrogen bonds (dashed lines) shown.

To test whether the TnsB–TnsC interactions that we observe in the transpososome structure are required for the ability of TnsB to stimulate the ATPase activity of TnsC (and trimming of the TnsC filament), we tested various mutations using a malachite green ATP hydrolysis assay (Fig. 3e and Methods). Consistent with expectations, TnsC has low basal levels of ATPase activity on its own, and wild-type TnsB substantially stimulates ATPase activity (Fig. 3e). Introduction of the Y439A mutation in TnsB reduced the ability of TnsB to stimulate ATP hydrolysis, but other mutations (such as R432A; Fig. 3d,e) had no effect. To test whether domain IIβ is generally required, we next tested various TnsB fragments for their ability to stimulate ATP hydrolysis. The C-terminal domain (CTD, residues 476–584) of TnsB or a C-terminal truncation (ΔCTD, residues 1–475) of TnsB did not stimulate ATP hydrolysis, consistent with previous work[17]. By contrast, the C-terminal domain with the domain IIβ (IIβ+CTD, residues 410–584) increased ATP hydrolysis above basal levels of TnsC ATPase activity (that is, without TnsB) or CTD (Fig. 3e). Despite these effects on ATP hydrolysis, TnsB mutations at this interface only slightly reduced transposition activity compared with the wild type (Fig. 3f). However, consistent with the idea that these TnsB mutants have a reduced capacity to stimulate ATP hydrolysis, targeted transposition (as assessed by Illumina sequencing (Methods)) was significantly reduced in the absence of S15 (69% for the wild type versus less than 6% for the mutants) (Fig. 3f and Extended Data Fig. 6). The addition of S15, which has been shown to boost overall transposition activity[23], resulted in a less marked effect on on-site targeting (99% for the wild type versus 76–93% for the mutants) (Fig. 3f and Extended Data Fig. 6), consistent with the idea that S15 can generally promote targeted transposition by stabilizing CRISPR effector binding. It is particularly noteworthy that, in the absence of S15, wild-type TnsB retains high levels of on-site targeting (69%), whereas TnsB mutants have no targeting ability (Extended Data Fig. 6). These marked effects suggest that the ability of TnsB to promote TnsC filament disassembly is compromised, similar to the effects observed with AMP-PNP[18].

In this context, the defined assembly of TnsC that we observe in the transpososome may correspond to either an ATP hydrolysis-resistant state or a stable post-hydrolysis state. To distinguish between these two possibilities, we used local refinement (focusing on TnsC) to generate a 3.2 Å map (Supplementary Fig. 1c,d), that is sufficient to unambiguously distinguish between ATP and ADP. This improved map demonstrates that ATP is bound to all protomers (Fig. 3g and Supplementary Fig. 5). Our observations are consistent with the idea that although TnsB stimulates ATP hydrolysis and TnsC filament disassembly, aspects of the transpososome structure probably prevent hydrolysis, allowing the stable configurations observed here.

## Transpososome TnsC–DNA interactions

In the absence of additional factors, ATP-bound TnsC forms continuous helical filaments (referred to throughout as helical TnsC) with DNA, and has no preference for filament length[15,18]. However, the precise insertion profiles of *Sh*CAST suggest that TnsB trims back TnsC filaments to a specific length before transposon DNA integration[2]. Thus, we explored whether structural differences distinguish helical TnsC from the structure observed here. Although TnsC in the transpososome is globally similar to helical TnsC (Protein Data Bank (PDB) ID: 7M99; overall Cα r.m.s.d. = 1.6 Å), it is not helically symmetric (Supplementary Video 2 and Supplementary Fig. 6), because imposing helical symmetry resulted in a lower resolution reconstruction with aberrant features (Supplementary Fig. 7). The largest structural changes in the assembly correspond to the TnsB-proximal (TnsC1) and Cas12k-proximal (TnsC12) (Fig. 4a) TnsC protomers. These protomers are collapsed towards the centre of the TnsC assembly with Cα r.m.s.d. between 2.1 and 2.5 Å (Fig. 4a). This suggests that the slight conformational changes observed here are due to interactions with other CAST components within the transpososome, namely TniQ and TnsB.

Given the lack of structural rearrangements of TnsC within the transpososome, it was notable that TnsC–DNA interactions appeared

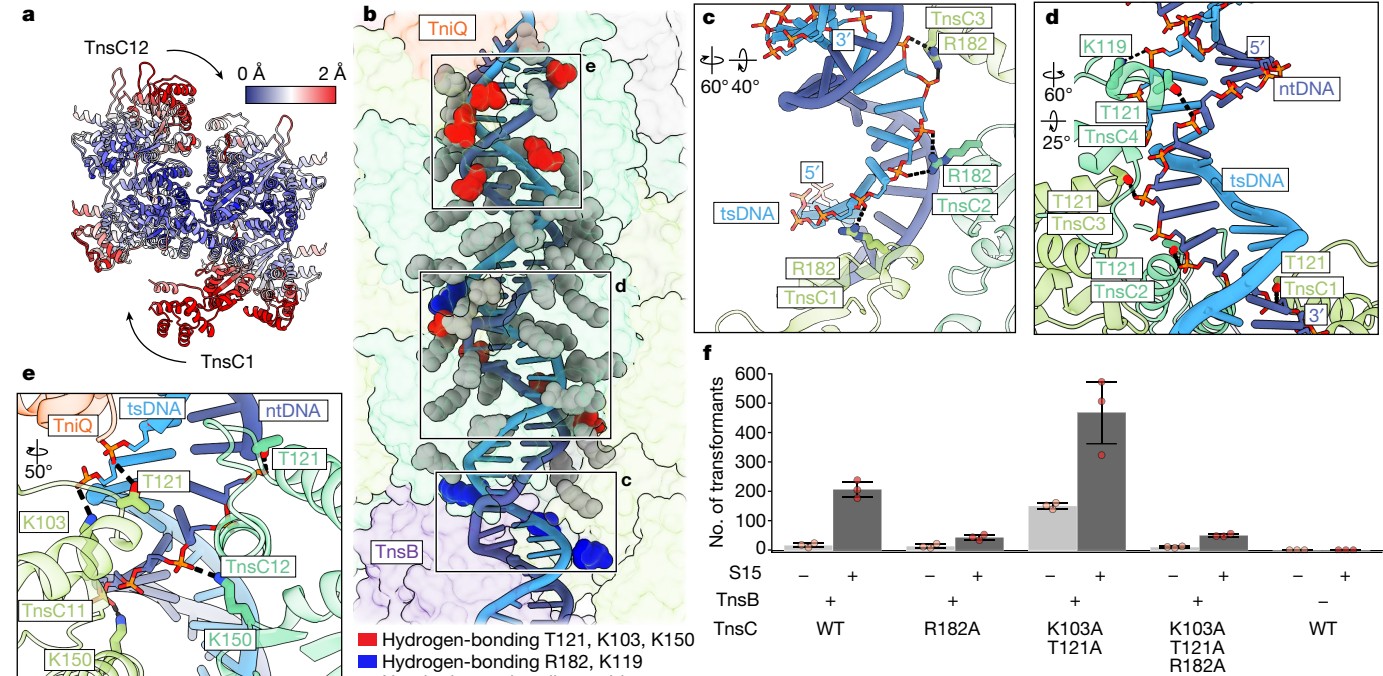

**Fig. 4 | TnsC forms a network of important interactions with target DNA throughout the transpososome. a**, TnsC protomers are coloured by changes in Cα r.m.s.d. between the helical TnsC (PDB: 7M99) and TnsC in the transpososome. The arrows represent the inward movement of TnsC, starting from the helical filament. **b**, TnsC–DNA interactions throughout the transpososome. TnsC residues forming hydrogen-bonding interactions with DNA, at least 4 Å away, are represented by red (K103, T121 and K150, previously identified interacting residues) or blue (R182 and K119; newly identified interacting residues) spheres. TnsC residues that are close to, but do not form specific interactions with the DNA backbone, more than 4 Å away, are represented by grey spheres. The target strand DNA (tsDNA) (light blue) and non-target strand DNA (ntDNA) (dark blue) are represented in ribbon. **c**, TnsB-proximal TnsC protomers interact with the sugar-phosphate backbone of tsDNA. **d**, TnsB-proximal TnsC protomers are shown to interact with the ntDNA backbone. **e**, TnsC protomers adjacent to Cas12k interact with both tsDNA and ntDNA immediately downstream of the R-loop. In **c**–**e**, hydrogen-bonding interactions between protein residues and the sugar-phosphate backbone of DNA are represented with dashed lines. Rotations relative to **b** are indicated. **f**, In vitro transposition assay of TnsC residues observed to interact with DNA. Data are mean ± s.d. (*n* = 3 biological triplicates). Raw data points are shown in red.

---

markedly different in the context of the transpososome compared with previous TnsC structures obtained in the absence of other transposition factors. In previous work, the helical symmetry of TnsC tracked with that of the bound DNA duplex[15,18]. In doing so, helical TnsC imposes its own helical symmetry, which distorts the bound DNA through underwinding[15,18]. In contrast to previous TnsC structures, within the transpososome, TnsC does not track specifically with either strand of the DNA duplex (Fig. 4b). Comparing the structure of transpososome target DNA to B-form DNA, we find that target DNA is only slightly underwound compared with B-form DNA (11 bp versus 10 bp per turn) and does not match the layer line spacing of TnsC in the transpososome (~40 Å; Extended Data Fig. 7).

Owing to the mismatch in TnsC (40 Å) and DNA (36 Å) repeat lengths, the specific TnsC–DNA interactions made in each of the two TnsC turns in the transpososome are distinct (Fig. 4b). In helical TnsC structures, residues K103 and T121 are consistently found to track with one particular strand of duplex DNA[18] (5′ to 3′ in the direction of the C-terminal to the N-terminal face, Supplementary Fig. 8), which we confirmed in improved, high-resolution (3.5 Å) cryo-EM reconstructions of TniQ–TnsC (Supplementary Fig. 9 and 10). In the context of the transpososome, we observed a network of interactions, some of which correspond to new interactions by previously identified DNA-binding residues[15,18] (T121, K103 and K150, red) (Fig. 4b) or new interactions made by residues not previously associating with DNA (K119 and R182, blue) (Fig. 4b). The remaining interactions throughout the two turns of TnsC correspond to general electrostatic interactions, as these residues are too distant to form specific hydrogen-bonding interactions with the DNA backbone (grey residues) (Fig. 4b and Extended Data Table 2).

Notably, we find that the two different sets of residues forming specific interactions with the DNA backbone are generally segregated spatially, either towards the Cas12k-proximal end (red residues) (Fig. 4b) or towards the TnsB-proximal end (blue residues) (Fig. 4b). In particular, the new interactions with the DNA backbone, involving R182 (Fig. 4c) and K119 (Fig. 4d) are supported by cryo-EM density (Supplementary Fig. 11). Residues previously shown to interact with DNA (T121 and K103) in previous structures of TnsC are now shown to associate with both target and non-target strands of the DNA duplex (Fig. 4e).

We thus hypothesized that TnsC residue R182 is critical for *Sh*CAST transposition. Consistently, both single (R182A) and triple mutants (K103A/T121A/R182A) significantly decreased the overall transposition activity (Fig. 4f), whereas the double mutant K103/T121 showed an overall increase in transposition activity (Fig. 4f), consistent with our previous results[18]. Mutant phenotypes were consistent regardless of whether S15 is added (Fig. 4f), further emphasizing the importance of R182. We hypothesize that this network of interactions corresponds to the distinct roles of TnsC residues in transpososome formation and stabilization. Furthermore, because these interactions are not observed in the TniQ–TnsC structure (Supplementary Fig. 10), this is not solely owing to the binding of TniQ, but must be related to R-loop formation, consistent with other reports[23].

## The basis of insertion spacing variability

We were able to distinguish two different transpososome populations containing either 12 or 13 TnsC protomers (the former is shown in Fig. 1). Because particles containing 12 TnsC protomers were most

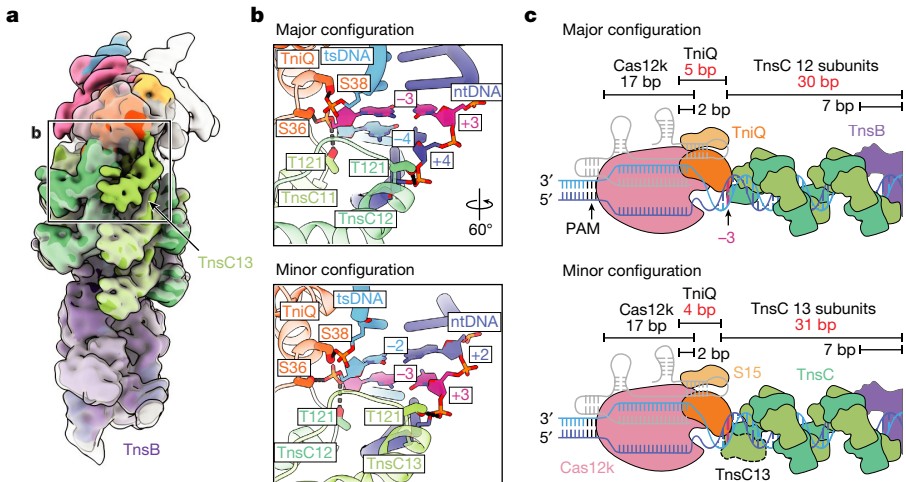

**Fig. 5 | Two different transpososome configurations differ at the Cas12k-proximal end. a**, The cryo-EM reconstructions (low pass filtered to 10Å) of major and minor configurations of transpososome. The reconstructions were aligned with respect to TnsB, where protein–DNA interactions are identical in the two configurations. The map corresponding to the minor configuration is coloured and opaque, and the major configuration is transparent. The outline indicates TnsC–DNA interactions at the Cas12k face. **b**, Expanded view of the outlined region in **a**, showing that major and minor configurations interact with both target-strand DNA (tsDNA) and non-target strand DNA (ntDNA). Residues from TniQ and TnsC in the minor configuration interact with the phosphate backbone that are shifter by 1 bp towards the R-loop compared with the major configuration. Hydrogen-bonding interactions between protein residues and the DNA backbone are represented with dashed lines.

Base pairs interacting with the final two protomers of TnsC are represented in stick. Positions three bases downstream of the R-loop (−3 on tsDNA and +3 on ntDNA) are coloured magenta as a landmark, facilitating comparison of the two configurations. **c**, Schematic of DNA-binding contacts of each component (Cas12k, TniQ and TnsC) in major (top) and minor (bottom) configurations. The number of base pairs contacted by each component is shown and the length is indicated with bar-ended lines. Positions interacting with two protein components (Cas12k and TniQ or TnsC and TnsB) are indicated as overlapping bars. The minor configuration includes and additional TnsC protomer (TnsC13, dashed outline) proximal to Cas12k, which makes the DNA-binding contribution of TniQ and TnsC different in the two configurations (highlighted in red). The third base pair downstream of the R-loop is coloured magenta, as in **b**.

frequently observed (73% of particles used in final refinement) and yielded a slightly higher resolution cryo-EM map (3.5 Å versus 3.7 Å; Extended Data Table 1), we refer to this configuration as the major configuration. Comparing the two configurations, the specific TnsC–DNA interactions in protomers adjacent to TnsB are virtually identical, since the same interactions are made at the same positions on the target DNA substrate (Extended Data Fig. 8). Aligning both transpososome reconstructions (major and minor configurations) on the basis of TnsB reveals an additional TnsC protomer (TnsC13) next to Cas12k in the minor configuration, which contains an additional TnsC protomer compared with the major configuration (12 TnsC protomers) (Fig. 5a). Correspondingly, Cas12k is rotated by approximately 60° in the minor configuration (solid surface, Fig. 5a) compared with the major configuration (transparent surface, Fig. 5a). At first glance, the TnsC–DNA contacts adjacent to Cas12k also appear similar. However, these contacts are shifted over by one nucleotide in the minor configuration compared with the major configuration (Fig. 5b). That is, the last two TnsC protomers in both configurations interact in the same way with TniQ, resulting in a translational shift of TnsC–DNA interactions (Fig. 5b,c). This leads to slight variability in the number of base pairs contacted by TniQ (5 bp (major configuration) versus 4 (minor configuration)) (Fig. 5c) and TnsC (30 bp versus 31 bp, respectively) (Fig. 5c). The structural variations described here suggest that more than one transpososome complex is stable, leading to an elegant mechanism to explain the slight variability (around 5 bp) in *Sh*CAST insertion profiles[2].

## Discussion

Our structural observations enable us to fill in crucial mechanistic details of the recruitment and insertion of transposon DNA at a CRISPR-defined target site (Fig. 6). Once a productive complex is formed consisting of target-site proteins Cas12k, S15, TniQ and TnsC

(Fig. 6a,b), TnsB–TnsC interactions stimulate ATP hydrolysis and TnsC filament disassembly. Our structure also suggests that the TnsB–TnsC contacts required for filament disassembly involve two TnsB domains, TnsB[hook] and domain IIβ (Fig. 6c). ATP hydrolysis (and TnsC filament disassembly) continues until TnsB encounters the two TnsC turns closest to Cas12k. We hypothesize that two turns of TnsC are required for transpososome formation with the *Sh*CAST element (Fig. 6d), a state resistant to TnsB-mediated ATP hydrolysis (Fig. 6e) and stabilized by new TnsC–DNA contacts, among them K103, T121, K119 and R182. These contacts appear to stabilize a TnsC configuration that is not fully engaged with DNA compared to the helical TnsC formed outside the transpososome. We speculate that the two turns of TnsC that resist TnsB-promoted disassembly might be a basis for the specific activation of TnsB at the target site. However, the detailed mechanism whereby TnsC activates ATPase activity, resists filament disassembly and stimulates integration requires further investigation.

This structure also reveals how *Sh*CAST transpososome architecture conserves TnsC function across Tn7-like elements. *Sh*CAST TnsC forms continuous helical filaments on DNA[15,18], which are distinct from TnsC configurations found in the prototypic Tn7[19] and I-F3 CAST subfamily[11]. TnsC from prototypic Tn7 forms heptameric oligomers[19] and led to the original hypothesis that the ring would allow segregated interaction interfaces: the TnsD interface maps to the N-terminal face, and the TnsA–TnsB interface maps to the C-terminal face[19]. Although the I-F3a CAST TnsC adopts a heptameric structure[11], it remains unknown whether TnsC in this system has similarly segregated interaction interfaces. However, chromatin immunoprecipitation–sequencing experiments suggest that TnsC has a key role in associating with the Cascade complex, serving as a selectivity factor in subsequent recruitment of TnsA–TnsB[11]. Therefore, despite the structural differences in *Sh*CAST TnsC, we hypothesize that the segregated interactions that we observe in the *Sh*CAST transpososome are conserved across Tn7-like transposition systems.

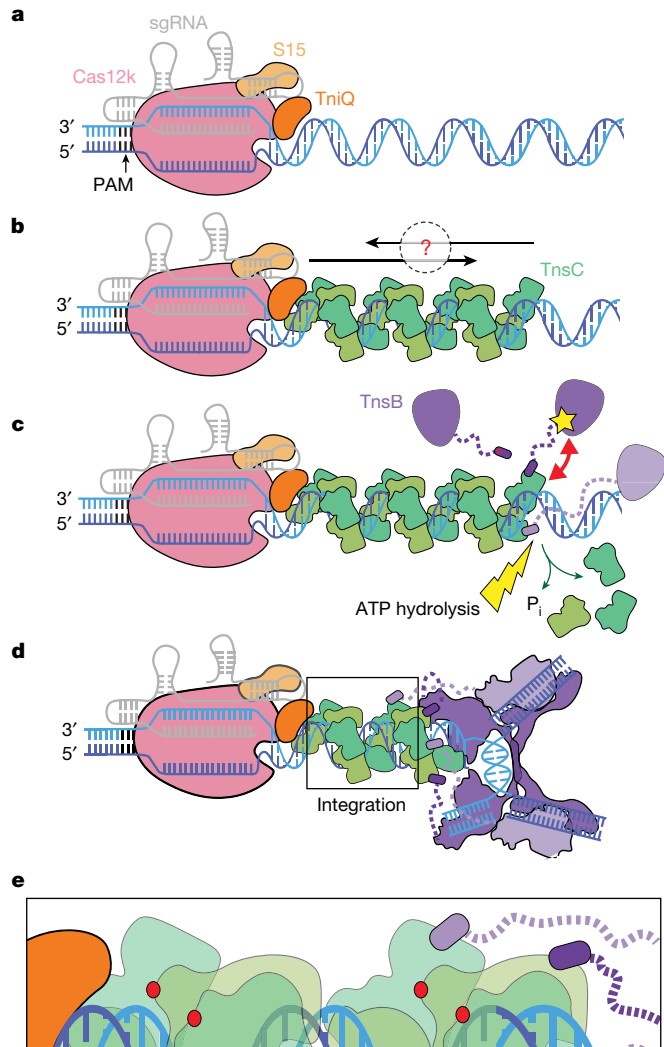

The segregation of interactions that we observe in the transposome also suggests a mechanism for how the core transposition machinery can co-opt different CRISPR effectors via TniQ. This is particularly interesting, considering the I-B CAST elements, which contain features of both prototypic Tn7 and CAST elements[6]. TnsC and the cognate transposase in I-B CAST elements can associate with two different TniQ domain proteins: TnsD (sequence-specific DNA-binding protein) or TniQ (associated with a CRISPR effector) to direct insertions[6]. Competition between these two targeting modalities suggests that a single interface on TnsC can interact with either TniQ or TnsD[6]. Our structure provides an elegant explanation for how this is achieved: by separating targeting machinery (TniQ or TnsD) from the core transposition machinery (TnsA–TnsB) via segregated interfaces on TnsC. This ensures that co-option of target-site-binding proteins (that is, with different CRISPR effectors or different DNA-binding proteins) does not interfere with the function of core transposition machinery.

Although the transpososome structure is remarkably uniform and well-resolved (Extended Data Fig. 2) compared with the heterogeneity of other reconstitutions[23], we observe a slight variability in the oligomeric composition of TnsC within transpososome populations. This is not altogether unexpected, since CAST insertion profiles also contain a slight variability (around 5 bp) in their insertion profiles[2]. Notably, the protein–protein associations that we observe in either configuration are essentially identical, except that TnsC and TniQ are shifted one nucleotide closer to Cas12k (Fig. 5c). Therefore, transpososome associations are constructed to allow minor flexibility. Taking this a step further, it is possible that the basis of CAST insertion variability lies in the stable configurations of TnsC that preserve CAST component interactions. This suggests that more precise CAST elements can be engineered by reinforcing (that is, stabilizing) specific spatial associations between CAST components within the transpososome.

## Online content

**Fig. 6 | Mechanistic model of recruitment and integration of the transposition components in *Sh*CAST. a**, The CRISPR effector Cas12k (pink) bound to sgRNA (grey) defines the target site by forming an R-loop and associating with S15 (tan) and TniQ (orange). **b**, TnsC (green) may polymerize on DNA (blue) in two directions: towards or away from the target site (indicated by the black arrows and the question mark). TnsC interacts with target-site-associated proteins (Cas12k–sgRNA, S15 and TniQ) to form the recruitment complex. **c**, Two interactions between TnsB (purple) and TnsC: TnsB is recruited to TnsC filaments by TnsB[hook] (purple, indicated with a red asterisk) and domain IIβ (indicated approximately by a yellow star) from TnsB can interact with TnsC (indicated by a red double arrow), lead to disassembly of TnsC and promote ATP hydrolysis (yellow lightning bolt), resulting in the release of phosphate (green arrows). Dashed lines (purple) represent flexible linkers between TnsB[hook] and the rest of TnsB. **d**, TnsB is recruited to the target site and forms the STC upon integration. Four TnsB[hook] are bound to four TnsC protomers proximal to TnsB through the flexible linker (dashed purple line). Two turns of TnsC (12 protomers) are stably formed against disassembly. Together with target-site-associated proteins and TnsB STC, all CAST components form the transpososome at the integration site. **e**, A magnified view of TnsC (boxed) in the transpososome shown in **d**. All TnsC protomers are in the ATP-bound state (ATP is shown as a red circle). TnsC in the transpososome does not specifically track with DNA helical symmetry, unlike previous helical structures of TnsC.

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

## Methods

### Protein cloning of *Sh*CAST mutants and S15

TnsC (K103A, T121A, and R182A), TnsB (ΔCTD; residues 1–475, CTD; residues 476–584, IIβ+CTD; residues 410–584, R432A, and Y439A), and TniQ (Δ1–10; residues 11–167, W10A, F12A, and H34A) mutants were cloned from pXT130-TwinStrep-SUMO-ShTnsC (Addgene #135526), pXT129-TwinStrep-SUMO-ShTnsB (Addgene #135525) and pXT131-TwinStrep-SUMO-ShTniQ (Addgene #135527), respectively, using the Q5 site-directed mutagenesis kit (NEB) and two corresponding primers (forward and reverse primer pair are listed in Supplementary Table 1). Double or triple mutants were generated by introducing mutation on top of the mutant plasmid vectors. For S15, the DNA sequence for *Escherichia coli* S15 was amplified by PCR from *E. coli* K-12 MG1655 chromosomal DNA using the following primers: RPS15_Ext_Hifi_F and RPS15_Ext_Hifi_R (Supplementary Table 1). This gene fragment was cloned into a linearized vector backbone (original ShTniQ vector from Addgene #135527) containing TwinStrep-SUMO at the N terminus of S15 using the NEBuilder HiFi assembly protocol (NEB). TnsC clones were transformed into *E. coli* BL21(DE3) competent cells (NEB), while TnsB and S15 clones were transformed into homemade T1 phage-resistant pRIPL BL21(DE3), derived from transforming pRIPL into BL21(DE3) Competent *E. coli* (NEB).

### Protein expression and purification of *Sh*CAST components and S15

All clones except S15 were purified using previously described protocols[2,18]. For purification of S15, single colonies were inoculated in 10 ml LB medium containing 34 μg ml⁻¹ chloramphenicol and 100 μg ml⁻¹ ampicillin, and grown overnight at 37 °C as starter cultures. The overnight starter culture was added to 1.5 L of TB medium containing the same ratio of antibiotics and grown until an absorbance of 0.4 at 37 °C. The temperature was lowered to 20 °C and the culture was induced at an absorbance of 0.6 with 0.4 mM isopropyl-β-ᴅ-thiogalactopyranoside and continued to grow overnight (16–18 h). Cells were collected by centrifugation at 5,000 rpm for 10 min and resuspended in lysis buffer (50 mM Tris pH 7.4, 500 mM NaCl, 5% glycerol, 1 mM DTT, 1 mM PMSF, 0.1% NP-40 and protease inhibitors (Pierce, ThermoScientific)). The cell lysate was sonicated for 15 min with a pulse on for 2 s with 10 s rest in between. The sonicated lysate was centrifuged for 30 min at 10,000 rpm at 4 °C. The supernatant was loaded onto a Strep-Tactin Superflow resin column (Qiagen) equilibrated with loading buffer (50 mM Tris pH 7.4, 500 mM NaCl, 1 mM DTT, and 5% glycerol). The column was washed with 5 column volumes (~15 ml) of wash buffer (50 mM Tris pH 7.4, 500 mM NaCl, 5% glycerol) before eluting with elution buffer (50 mM Tris pH 7.4, 500 mM NaCl, 5% glycerol and 4 mM desthiobiotin (SigmaAldrich)). The consolidated eluate was then incubated overnight at 4 °C with SUMO protease (1/100 mass ratio of SUMO protease to S15 protein).

S15 was next purified using a heparin purification step. Protein from the Strep-Tactin elution (in 500mM NaCl) was diluted to a final salt concentration of 200 mM NaCl before loading onto a 1 ml heparin column (GE Healthcare) equilibrated in low-salt buffer (50 mM Tris pH 7.4, 200 mM NaCl, 1 mM DTT and 5% glycerol). The heparin column was washed with 20 column volumes of low-salt buffer, followed by a gradient elution from 200mM NaCl to 1 M NaCl using high-salt buffer (50 mM Tris pH 7.4, 1.2M NaCl, 1 mM DTT, and 5% glycerol). Fractions containing S15 were concentrated, and buffer exchanged before size-exclusion chromatography was performed using a Superdex 200 increase 10/300 (Cytiva) equilibrated in the following buffer: 20 mM HEPES pH 7.5, 250 mM KCl, 1 mM DTT, and 5% glycerol. Fractions containing S15 were consolidated and concentrated to ~2 mg ml⁻¹ using a 6 ml 3K molecular weight cut-off centrifugal concentrator (ThermoFisher), before being aliquoted and flash-frozen in liquid nitrogen for long-term storage at −80 °C.

### Preparation of DNA substrate

The general idea of the substrate design for the STC was described previously[17]. However, instead of a symmetric strand-transfer DNA, we designed an asymmetric DNA that mimics the Cas12k-guided transposition product (Supplementary Table 1). This design includes two DNA substrates: (1) target-pot DNA containing left-end (LE) sequence and protospacer and (2) donor pot DNA containing right-end (RE) sequence. Each DNA substrate was prepared independently. First, target-pot DNA substrate was prepared by annealing four synthetic oligonucleotides (IDT) of the following: target-LE_F, non-target_R, LE_R, and 5′ dethiobiotinylated LUEGO[29] (Supplementary Table 1). These four oligonucleotides were mixed in a molar ratio of 10:11:11:10. Second, donor pot DNA is composed of three synthetic oligonucleotides including: RE_F, RE_R1 and RE_R2 (Supplementary Table 1). These oligonucleotides were mixed in a 10:11:11 molar ratio. The mixture of oligonucleotides was supplemented with a 10× concentrated annealing buffer to make the final buffer composition the following: 10 mM Tris pH 7.5, 50 mM NaCl, and 1 mM EDTA. The mixture was then heated up to 95 °C for 5 min and cooled down to 30 °C at the rate of 1 °C per minute using a thermal cycler (BioRad).

### Preparation of sgRNA

Using two primers, sgRNA_For and sgRNA_PSP1_Rev (Supplementary Table 1), the DNA sequence encoding the T7 RNA polymerase promoter and sgRNA was amplified by PCR from pHelper_ShCAST_sgRNA vector (Addgene, #127921). The DNA template was then subjected to GeneJet PCR purification (ThermoScientific). sgRNA was produced by in vitro transcription using the HiScribe T7 High Yield RNA synthesis kit (NEB) with the PCR amplified DNA sequence as the template. Purified sgRNA was aliquoted and stored at −20 °C.

### Reconstitution of transpososome complex and recruitment complex

The transpososome complex and recruitment complex were generated by stepwise assembly using a pull-down assay on streptavidin beads (Fig. 1b). For the target pot, annealed target-pot DNA was first bound to Streptavidin Mag Sepharose magnetic beads (Cytiva) equilibrated with reaction buffer (20 mM HEPES pH 7.5, 250 mM KCl, 15 mM MgCl₂, and 1 mM DTT) and incubated at room temperature with shaking for 30 min. Cas12k and sgRNA were reconstituted in a 1:2 molar ratio and incubated for 30 min at 37 °C in the following buffer: 20 mM HEPES pH 7.5, 160 mM NaCl, 15 mM MgCl₂, and 1 mM DTT. The Cas12k–sgRNA complex was added to the target-pot DNA-bound beads and incubated for 30 min at 37 °C in the presence of a fivefold molar excess of S15 to DNA. The beads were washed three times with 500 μl wash buffer (20 mM HEPES pH 7.5, 250 mM KCl, 15 mM MgCl₂, 0.05% Tween20, and 1 mM DTT) to remove excess protein and nucleic acids. Twentyfold molar excess of TniQ and tenfold molar excess of S15 to DNA were added to the beads and incubated at 37 °C for 30 min. Tenfold molar excess of TnsC was diluted to a final salt concentration of 250 mM NaCl and added to the TniQ–S15 beads solution containing a final concentration of 1 mM ATP. The sample was incubated for 30 min at 37 °C before the beads were washed three times with 500 μl wash buffer containing 1 mM ATP. The recruitment complex was prepared by eluting the target-pot DNA at this point using 50 μl of elution buffer (20 mM HEPES, 250 mM KCl, 15 mM MgCl₂, 0.05% Tween20, 10 mM biotin, 1 mM ATP, and 1 mM DTT). For transpososome assembly, donor pot containing TnsB and donor pot DNA was prepared independently in 6:1 molar ratio with the following final buffer composition: 25 mM HEPES pH 7.5, 100 mM NaCl, 15 mM MgCl₂, and 1 mM DTT; and incubated at 37 °C for 30 min. The reconstituted donor pot was added to the beads and incubated for 40 min at 37 °C. Three washes of 500 μl wash buffer containing 1 mM ATP were performed before eluting the transpososome complex from the beads with 50 μl elution buffer. The eluate was diluted with the

wash buffer containing 1 mM ATP for cryo-EM or negative-staining EM sample preparation by 6–10 fold or 15 fold respectively.

## In vitro transposition assay and high-throughput mapping of transposition events

All purified proteins (except Cas12k) were diluted to 2.5 µM using stock buffer (500 mM NaCl, 50 mM Tris pH 7.4, 10% glycerol, 0.5 mM EDTA, 1 mM DTT). Cas12k was assembled with sgRNA before incubation with other components in the following manner: sgRNA was added to Cas12k to make 2.5 µM Cas12k and 30 µM sgRNA. Separate reactions were prepared for the target pot and donor pot. The target pot contains 2.24 nM pTarget_pBC_KS+_PSP1 (a gift from the J. E. Peters laboratory), 104 nM of Cas12k, 104 nM TnsC, 104 nM TniQ and 1.25 µM sgRNA in transposition buffer (26 mM HEPES, 50 mM KCl, 0.2 mM $MgCl_2$) supplemented with 2 mM DTT, 50 µg ml$^{-1}$ BSA and 2 mM ATP. The donor pot contains 1.08 nM pDonor_ShCAST_kanR (Addgene #127924), 104 nM TnsB, 2 mM DTT, 50 µg ml$^{-1}$ BSA and 2 mM ATP in transposition buffer. Both pots were incubated at 37 °C for 30 min before combining and adding MgOAc$_2$ to a final concentration of 15 mM. The combined reaction was further incubated at 37 °C for 2 h. After the final incubation step, 20 µl of the reaction mixture was digested by 1 µl of Proteinase K (Ambion) and incubated at 37 °C for 1 h. 10 µl of the reaction was then transformed into 100 µl of Stellar competent cells (Takara Bio) and plated on kanamycin plates. All the experiments were done in biological triplicates.

From the plates of each condition, the colonies were washed up and diluted to absorbance of 0.5 using LB. The liquid culture was then grown for 3 h at 37 °C before the plasmid extraction using a miniprep kit (QIAGEN). Prepared DNA was sequenced by the Microbial Genome Sequencing Center (https://www.seqcenter.com/) using the Illumina DNA sequencing service on the NextSeq 2000 platform. Paired-end reads (2 × 151 bp) were analysed using BBtools v38.98 (http://sourceforge.net/projects/bbmap/). First, the reads were processed by BBDuk (BBtools v38.98) to collect adjacent target DNA sequences from all the reads containing 30bp of the ShCAST left-end sequence. Then these sequences were mapped onto the pTarget_pBC_KS+_PSP1 using BBMap (BBtools v38.98) with a 90% sequence identity cut-off. Number of base pairs between the end of the PAM and the beginning of the ShCAST left-end sequence was used to define the position of the transposition. Based on the mapping, the transpositions at 55–70 bp downstream of the PAM were considered to be on-target transposition. The number of on-target transposition reads was divided by the total number of transposition reads to estimate the on-target ratio.

## Malachite green ATP hydrolysis assay

One-hundred millilitres of malachite green reagent was prepared by the following protocol. First, 34 mg of malachite green carbinol base (ChemCruz) was dissolved into 40 ml of 1N HCl. 1 g of Ammonium Molybdate (MacronChemicals) was dissolved in a separate 14 ml of 1N HCl. These two solutions were mixed and diluted up to 100 ml using ddH$_2$O. The reagent was then filtered using 0.45 µm syringe filter (VWR), and covered with an aluminum foil to avoid light. ATPase assay was completed using purified protein stocks of TnsC, and constructs of TnsB (wild type, ΔCTD, CTD, IIβ+CTD, R432A and Y439A). To minimize the processing time, 2× protein pot and 2× ATP pot were prepared independently. Buffer composition for both pots was the same as follows: 25 mM HEPES pH 7.5, 150 mM NaCl, 1 mM DTT. In addition to the described buffer, 2× protein pot contains 10 µM TnsC, 2 µM TnsB, and 2.5 µM 60 bp DNA (annealed using 60bp_top and 60bp_bot, Supplementary Table 1), and 2× ATP pot contains 1 mM ATP and 2 mM MgCl$_2$. One-hundred microlitres of each pot was mixed to initiate the ATP hydrolysis reaction. For each time point (10, 20, 30, 40, 50, 60, 90 and 120 min), 20 µl of the reaction mix was transferred to the 96-well plate (Corning) that contains 5 µl of 0.5 M EDTA to quench further hydrolysis. After taking all the time points, 150 µl of the room-temperature Malachite Green reagent was added to each well, and the plate was shaken

for ~5 min to develop the colour for imaging. Absorbance at 650 nm was read from each well using the i-control v1.101.4 software on an Infinite 200 plate reader (TECAN). For calibration, KH$_2$PO$_4$ solutions of the following concentrations were used to generate a calibration curve: 0 µM, 4 µM, 8 µM, 12 µM, 16 µM, 24 µM, 40 µM and 60 µM. The data was plotted with time as the x-axis and the concentration of released inorganic phosphate as the y-axis. The slope was obtained from the linear regression to get the reported $v_o$ values (in moleculas of ATP per min). Three independent experiments were done for each condition, to generate the reported bar plot.

## Cryo-EM sample preparation and freezing of transpososome complex

The transpososome complex sample was prepared by diluting the elution from the pull-down by 6-, 8- and 10-fold using wash buffer (described above) containing 1 mM ATP. Graphene oxide-coated grids were prepared following the protocol described previously[17,30,31]. Four microlitres of transpososome complex sample was loaded on the carbon side of a graphene oxide-coated grid and incubated for 20 s in the Mark IV vitrobot chamber (ThermoFisher), which was set to 4 °C and 100% humidity. Each grid was blotted for 6 s with a blot force of 5, and then plunged into liquid ethane cooled with liquid nitrogen.

## Cryo-EM imaging and image processing of transpososome complex

Vitrified samples of transpososome complex were first screened using Talos Arctica (ThermoFisher) operated at 200 kV prior to large-scale data collection. Screened grids were imaged using a Titan Krios (300 kV, ThermoFisher) equipped with a BioQuantum energy filter (Gatan) and K3 direct electron detector (Gatan). A total of 15,740 micrographs were collected using Leginon v3.5[32] at 81,000× magnification (1.067 Å per pixel) using image shift, with the nominal defocus from −0.8 µm to −2.5 µm. Each movie was collected with 2 s of exposure with a total dose of 49.91 electrons per Å$^2$, fractionated into 50 frames. Frames were aligned using MotionCor2[33] through Appion v3.4[34], which was then imported to cryoSPARC v3.3.1[35] for contrast transfer function (CTF) estimation and downstream image analysis. The workflow described below is shown in Extended Data Fig. 2. Using a template picker from cryoSPARC v3.3.1, initial particle stack was subjected to 2D classification in cryoSPARC v3.3.1. Two-dimensional classification resulted in 536,450 particles from selected 2D averages, which was subjected to ab initio reconstruction in cryoSPARC v3.3.1. Ab initio reconstruction separated particles into two classes, one containing 53% (285,090 particles) and another containing 47% (250,842 particles). The two classes were then subjected to homogenous refinement in cryoSPARC respectively. The particle stacks from both classes were subjected to 3D classification in RELION v4[36,37], which removed junk particles with weak densities of Cas12k or TnsB. Each classification resulted in a particle stack that had well-defined configurations of ShCAST transpososome. The final particle stacks (188,055 particles for major configuration of transpososome and 67,096 particles for minor configuration of transpososome) were subjected to iterative CTF refinements[38] and Bayesian polishing[39] to improve resolution. The final maps of both major and minor configurations were then subjected to local refinement in cryoSPARC v3.3.1. Volume maps of Cas12k–S15-TniQ only, TnsC only, and TnsB only were generated in UCSF Chimera v1.14[40] using the volume eraser tool to remove the map. A mask for Cas12k–S15–TniQ, TnsC and TnsB, respectively, was then generated in RELION v4[36] using the volume produced by Chimera as the input. Local refinement was then done in cryoSPARC v3.3.1[35] with the mask generated by RELION v4 and the full map without particle subtraction. Local refinement of the major configuration of the transpososome resulted in 3.1 Å for the Cas12k–S15–TniQ region, 3.2 Å for both the TnsC and TnsB region. Local refinement of the minor configuration of the transpososome resulted in 3.6 Å for the Cas12k–S15–TniQ region, 3.6 Å for TnsC, and 3.8 Å for the TnsB region.

For Fig. 2, the composite map was generated using UCSF Chimera v1.14[40] command 'vop maximum' with the aligned reconstructions from local refinements of Cas12k and TnsC. 3D Variability in cryoSPARC v3.3.1[41] was performed on the major configuration of the transpososome using the mask and particles from the final refinement. The filter resolution was set to 6 Å and the number of modes to solve was set to 3. After the job was completed, the 3D variability display job using the particles and volumes from the 3D Variability job was completed with the output mode set to simple with 20 frames. The first series was then visualized in UCSF Chimera v1.14[40] using the "vop morph" command with all 20 frames. The results are presented in Supplementary Video 1.

To impose helical symmetry on the two turns of TnsC in the transpososome, a mask was first created using a map of TnsC only (generated by the same protocol described above) as input in cryoSPARC v3.3.1[35]. Helical refinement was then done in cryoSPARC v3.3.1[35] with the mask and the full map (major configuration) without particle subtraction. Initial helical parameters of the ATPγS-bound TnsC filaments (PDB: 7M99, rise = 6.82 Å and twist = 60°)[18] were used. Helical parameters were refined with the following range: 6.14 Å to 7.50 Å for the helical rise, and from 57° to 63° for the twist.

## Model building of transpososome complex

Initial models from previous studies, including TnsC (PDB: 7M99)[18], TnsB (PDB: 7SVW)[17] and sgRNA (PDB: 7PLA)[15], were first docked into the cryo-EM density and manually rebuilt using Coot v0.9.8.2[42]. Additional models were created using AlphaFold2[43] for the following components: Cas12k, TniQ and S15, and rigid-body docked into the density. Coot v0.9.8.2 was used to manually remodel or rebuild sections of the model. Real space refinement of the DNA substrate was completed in Phenixv1.19.1-4122[44] with both base-pair and secondary structure restraints enforced. For measuring the interface area between TniQ and TnsC protomers, UCSF ChimeraXv1.2.5[45] command 'measure buriedarea' was used with the desired two chains as inputs.

## Model validation of transpososome complex

Map-model Fourier shell correlation (FSC) was computed using Mtriage in Phenixv1.19.1-4122[44]. Map-model FSC resolution of each dataset was estimated from Mtriage FSC curve, using 0.5 cut-off. The masked cross-correlation (CCmask) from Mtriage was reported as representative model-map cross-correlation. Model geometry was validated using MolProbity v4.5.1[46]. All the deposited models were submitted to MolProbity v4.5.1 server to check the clashes between atoms, Ramachandran plot, bond angles, bond lengths, side-chain rotamers, CaBLAM and C-beta outliers. All the model validation stats are summarized in Extended Data Table 1.

## Helical parameter estimation using Rosetta

Helical parameters are estimated between the adjacent TnsC protomers within the complexes of the following: ATPγS-bound TnsC helical filament (PDB: 7M99), major configuration, and minor configuration of the transpososome. For example, from the major configuration of transpososome, helical parameters were estimated in between TnsC1 and TnsC2, between TnsC2 and TnsC3, between TnsC3 and TnsC4, and so on. Rosetta tool 'make_symmdef_file.pl' was used for each pair of the TnsC protomers, with an example command presented at the bottom of this section. Obtained helical rises and turns were averaged to plot a bar-graph presented in Supplementary Fig. 6.

The Rosetta command that we used for helical parameter is Rosetta/main/source/src/apps/public/symmetry/make_symmdef_file.pl -m HELIX -p input.pdb -a A -b B.

## Reconstitution of TnsB^CTD–TnsC–TniQ complex and imaging for cryo-EM

The C-terminal 109 residues from wild-type TnsB (termed hereafter as TnsB^CTD) was cloned from pXT129_TwinStrep-SUMO-ShTnsB vector (Addgene, #135525) using Q5 site-directed mutagenesis kit (NEB) and two primers of following: TnsB_CTD_For and Ndel_primer_Rev (Supplementary Table 1). Cloned TnsB^CTD was transformed to E. coli BL21-RIPL competent cells (Agilent) and purified following the protocol for TniQ purification. DNA substrate was prepared by annealing two synthetic oligonucleotides of BCQ_top and BCQ_bot (Supplementary Table 1). TniQ and TnsC were buffer exchanged into 25 mM HEPES pH 7.5, 200 mM NaCl, 2% glycerol, and 1 mM DTT (reaction buffer) prior to reconstitution of the complex. TnsC filament was first reconstituted by supplementing the reaction buffer with the following components: 128 μM TnsC, 8 μM DNA, 2 mM ATP, and 2 mM MgCl$_2$. After 5 min of incubation on ice, sixfold molar excess of TniQ was added to TnsC filaments and incubated at 37 °C for 1 h. TnsB^CTD was then added to the TniQ–TnsC mixture (4:1 molar ratio of TnsB^CTD to TnsC), followed by incubation at 37 °C for 40 min. Four microlitres of the TnsB^CTD–TnsC–TniQ sample was loaded on UltrAuFoil R1.2/1.3 gold grids (Quantifoil) and vitrified using the Mark IV Vitrobot (ThermoFisher) set to 100% humidity and 4 °C. Samples were blotted for 7 s with blot force 5, and then plunged into liquid ethane cooled with liquid nitrogen. Vitrified grids were imaged using Talos Arctica (ThermoFisher, 200 kV) with K3 direct detector (Gatan) and BioQuantum energy filter (Gatan), at 63,000× nominal magnification (1.33Å per pixel). SerialEM v4.0[47] was used for data acquisition with 3 × 3 image shift, and nominal defocus range from −1 μm to −2.5 μm. Total dose was set to 50 e$^-$ Å$^{-2}$, which was fractionated into 50 frames.

## Image processing and model building for TnsB^CTD–TnsC–TniQ complex

Beam-induced motion correction, CTF estimation, and initial particle picking were done using Warp v1.0.9[48] for the collected 1,271 movies. Initial particle stack from Warp was subjected to 2D classification cryoSPARC v3.3.1[35], to get subset of TniQ-bound particles to train topaz neural network[49], which resulted in the 795,631 particles. 2D classification resulted in 624,597 particles from selected 2D averages with high-resolution features, which were subjected to heterogeneous refinement in cryoSPARC v3.3.1[35]. One class with better resolved TniQ (34%, 214,291 particles) was selected for downstream non-uniform refinement in cryoSPARC v3.3.1[35]. The particle stack was subjected to 3D classification in RELION v4[36,37] resulting in the intermediate particle stack with a stronger density of TniQ (70%, 150,358 particles). To improve the resolution of TniQ, this particle stack was subjected to two rounds of focused 3D classification (skipping alignments, tau fudge factor of 16), iterative CTF refinements[38] and bayesian polishing[39]. This resulted in the final dataset of 61,515 particles, with the gold standard FSC resolution of 3.5 Å. Due to the local variation of the map quality, LocSpiral[50] was used to post-process the map through COSMIC[2 51]. For model building of TnsC and TniQ, previously published structure of TnsC (PDB: 7M99)[18], and AlphaFold2[43] generated TniQ were manually docked into the density, followed by manual editing using Coot v0.9.8.2[42] and relaxation using Rosetta relax[52].

## Reporting summary

Further information on research design is available in the Nature Portfolio Reporting Summary linked to this article.

## Data availability

The atomic models are available through the PDB with accession codes: 8EA3 (major configuration), 8EA4 (minor configuration) and 7SVU (TnsB^CTD–TnsC–TniQ complex). All cryo-EM reconstructions are available through the EMDB with accession codes: EMD-27971 (major configuration), EMD-27972 (minor configuration) and EMD-25453 (TnsB^CTD–TnsC–TniQ complex).

29. Jullien, N. & Herman, J. P. LUEGO: a cost and time saving gel shift procedure. *Biotechniques* **51**, 267–269 (2011).

30. Wang, F. et al. General and robust covalently linked graphene oxide affinity grids for high-resolution cryo-EM. *Proc. Natl Acad. Sci. USA* **117**, 24269–24273 (2020).
31. Patel, A., Toso, D., Litvak, A. & Nogales, E. Efficient graphene oxide coating improves cryo-EM sample preparation and data collection from tilted grids. Preprint at *bioRxiv* https://doi.org/10.1101/2021.03.08.434344 (2021).
32. Suloway, C. et al. Automated molecular microscopy: the new Leginon system. *J. Struct. Biol.* **151**, 41–60 (2005).
33. Zheng, S. Q. et al. MotionCor2: anisotropic correction of beam-induced motion for improved cryo-electron microscopy. *Nat. Methods* **14**, 331–332 (2017).
34. Lander, G. C. et al. Appion: an integrated, database-driven pipeline to facilitate EM image processing. *J. Struct. Biol.* **166**, 95–102 (2009).
35. Punjani, A., Rubinstein, J. L., Fleet, D. J. & Brubaker, M. A. cryoSPARC: algorithms for rapid unsupervised cryo-EM structure determination. *Nat. Methods* **14**, 290–296 (2017).
36. Scheres, S. H. RELION: implementation of a Bayesian approach to cryo-EM structure determination. *J. Struct. Biol.* **180**, 519–530 (2012).
37. Scheres, S. H. Processing of structurally heterogeneous cryo-EM data in RELION. *Methods Enzymol.* **579**, 125–157 (2016).
38. Zivanov, J. et al. New tools for automated high-resolution cryo-EM structure determination in RELION-3. *eLife* **7**, e42166 (2018).
39. Zivanov, J., Nakane, T. & Scheres, S. H. W. A Bayesian approach to beam-induced motion correction in cryo-EM single-particle analysis. *IUCrJ* **6**, 5–17 (2019).
40. Pettersen, E. F. et al. UCSF Chimera-a visualization system for exploratory research and analysis. *J. Comput. Chem.* **25**, 1605–1612 (2004).
41. Punjani, A. & Fleet, D. J. 3D variability analysis: resolving continuous flexibility and discrete heterogeneity from single particle cryo-EM. *J. Struct. Biol.* **213**, 107702 (2021).
42. Emsley, P., Lohkamp, B., Scott, W. G. & Cowtan, K. Features and development of Coot. *Acta Crystallogr. D* **66**, 486–501 (2010).
43. Jumper, J. et al. Highly accurate protein structure prediction with AlphaFold. *Nature* **596**, 583–589 (2021).
44. Afonine, P. V. et al. New tools for the analysis and validation of cryo-EM maps and atomic models. *Acta Crystallogr. D* **74**, 814–840 (2018).
45. Goddard, T. D. et al. UCSF ChimeraX: meeting modern challenges in visualization and analysis. *Protein Sci.* **27**, 14–25 (2018).
46. Williams, C. J. et al. MolProbity: more and better reference data for improved all-atom structure validation. *Protein Sci.* **27**, 293–315 (2018).
47. Mastronarde, D. N. Automated electron microscope tomography using robust prediction of specimen movements. *J. Struct. Biol.* **152**, 36–51 (2005).
48. Tegunov, D. & Cramer, P. Real-time cryo-electron microscopy data preprocessing with Warp. *Nat. Methods* **16**, 1146–1152 (2019).
49. Bepler, T. et al. Positive-unlabeled convolutional neural networks for particle picking in cryo-electron micrographs. *Nat. Methods* **16**, 1153–1160 (2019).
50. Kaur, S. et al. Local computational methods to improve the interpretability and analysis of cryo-EM maps. *Nat. Commun.* **12**, 1240 (2021).
51. Cianfrocco MA, W. M., Youn, C., Wagner, R. & Leschziner, A. E. COSMIC²: a science gateway for cryo-electron microscopy structure determination. *Pract. Exp. Adv. Res. Comput.* http://doi.acm.org/10.1145/3093338.3093390 (2017).
52. Leaver-Fay, A. et al. ROSETTA3: an object-oriented software suite for the simulation and design of macromolecules. *Methods Enzymol.* **487**, 545–574 (2011).

**Acknowledgements** This work was performed at the National Center for Cryo-EM Access and Training (NCCAT) and the Simons Electron Microscopy Center located at the New York Structural Biology Center, supported by the NIH Common Fund Transformative High Resolution Cryo-Electron Microscopy Program (U24 GM129539,) and by grants from the Simons Foundation (SF349247) and NY State Assembly. This work also relied on data collected using an instrument supported by the NIH through award S10OD030470-01. We additionally acknowledge XSEDE for computational resources used for image processing (MCB200090 to E.H.K.). We thank S.-C. Hsieh for sharing the protocol to analyse Illumina sequencing results; J. E. Peters, S. H. Sternberg and A. Guarné along with the A. Guarné, J. E. Peters and E. H. Kellogg laboratories for feedback and stimulating discussion. This research is supported by the NIH: R01GM144566 (E.H.K.) and Pew Biomedical Foundation (E.H.K.). This work relied on data collected using an instrument supported by the NIH through award S10OD030470-01.

**Author contributions** E.H.K. designed and supervised the project. J.-U.P., A.W.-L.T. and T.X.W. carried out protein expression and purification. R.D.S. cloned S15. J.-U.P., A.W.-L.T. and V.H.T prepared sample components and completed DNA-based pulldowns. J.-U.P., A.W.-L.T., V.H.T., A.N.R. and T.X.W. prepared GO grids and optimized the sample for cryo-EM imaging. J.-U.P., A.W.-L.T. and A.N.R. collected, processed, analysed, and refined cryo-EM data. J.-U.P. and A.N.R. built and refined atomic models. J.-U.P. and A.W.-L.T. carried out all in vitro assays (ATP hydrolysis and in vitro transposition assays). J.-U.P., A.W.-L.T., A.N.R. and E.H.K. contributed to figures and manuscript writing. All authors contributed to manuscript writing and editing.

**Competing interests** The authors declare no competing interests.

**Additional information**
**Correspondence and requests for materials** should be addressed to Elizabeth H. Kellogg.

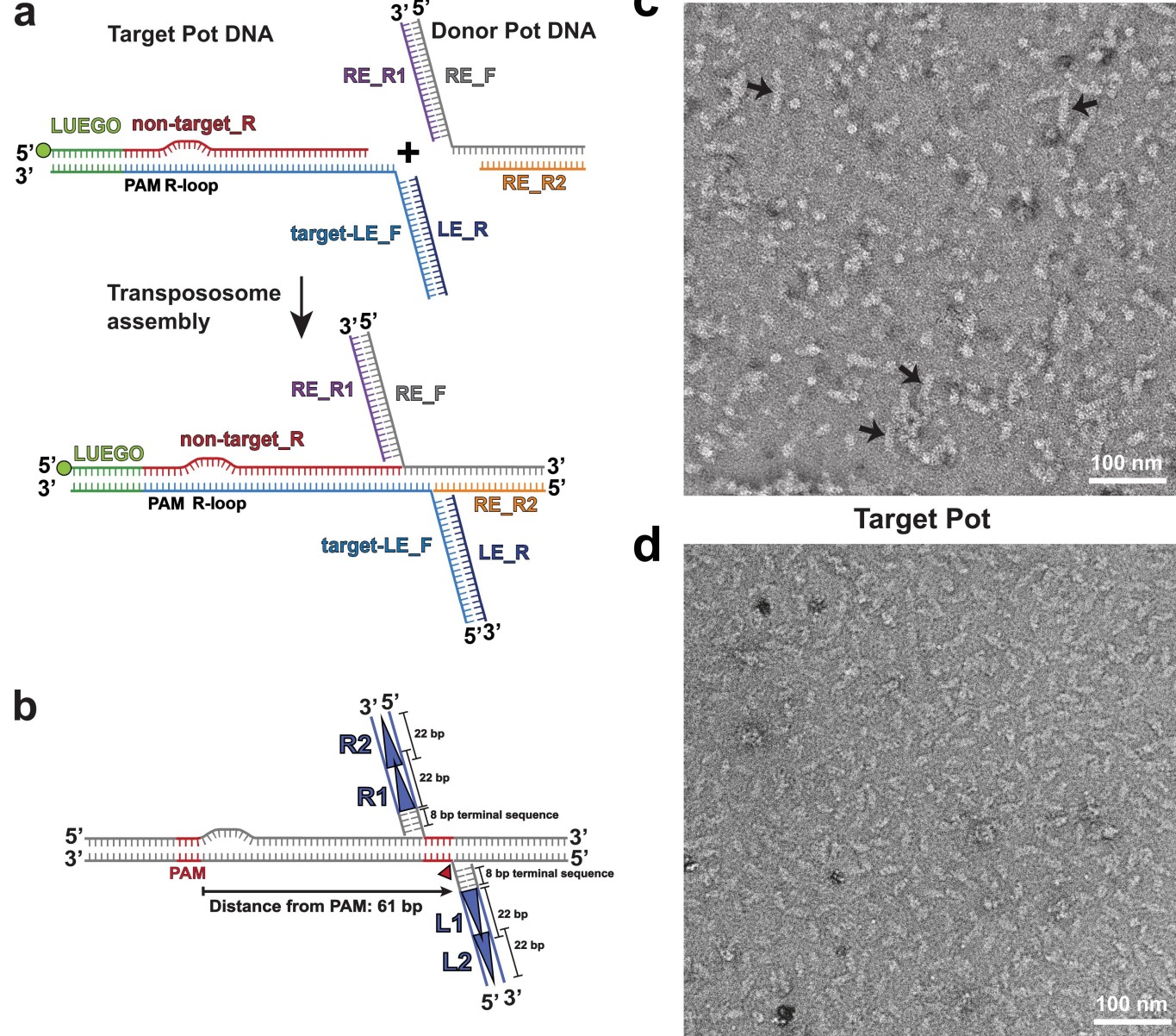

**Target Pot**

**Transpososome**

**Extended Data Fig. 1 | Designed DNA substrate schematic and negative staining microscopy of target pot and transpososome. a**. DNA substrate design for the transpososome assembly includes two DNA substrates: target pot DNA containing a Cas12k-binding site and the first two TnsB binding sites of left-end (LE), and donor pot DNA with the first two TnsB binding sites of right-end (RE). DNA substrates were designed to form a strand-transfer complex by having 5 base pairs (bp) single-stranded DNA (ssDNA) connecting the target DNA region and the transposon DNA region. The design for target-pot DNA is composed of four single-stranded DNA (ssDNA) of the following: Target-LE_F (light blue), non-target_R (red), desthiobiotinylated LUEGO (green), and LE_R (dark blue). Cas12k-binding region contains PAM and a 10 bp mismatch to facilitate R-loop formation as indicated as a displaced strand. 5′ end labeled desthiobiotin (green circle) on the LUEGO was used to conjugate target-pot DNA on magnetic beads for the pulldown. Second, donor-pot DNA consists of three ssDNA: RE_F (grey), RE_R1 (purple), and RE_R2 (orange). LE_R region and RE_R1 region of each DNA substrate corresponds to the first two TnsB binding sites of LE and RE, respectively. Two DNA substrates have 5 bp of complementary sequences to each other, which are annealed upon transpososome assembly. Locations of PAM, and R-loop are annotated in black. Sequences for all DNA substrates are included in Supplementary Table 1. **b**. Features of the designed DNA substrate. Each left-end and right-end transposon region of the DNA includes 8 bp terminal sequences and the two TnsB binding sites (L1 and L2 for the left end, R1 and R2 for the right end). The beginning of the left-end sequence (red triangle) is 61 bp distant from the PAM as indicated with a black arrow. PAM and the 5 bp of complementary sequences (target site duplication) were represented in red. **c**. Negative stain image of the target pot (i.e. recruitment complex containing Cas12k, S15, TniQ, and TnsC) shows a heterogenous assembly with variable length TnsC filaments (indicated by black arrows). **d**. Negative stain image of the transpososome complex shows the addition of donor pot disassembled TnsC filaments and resulted in a homogeneous sample. Scale bar (white) represents 100 nm. The micrographs shown are examples images from negative-stain screening datasets consisting of 20 and 100 micrographs, respectively.

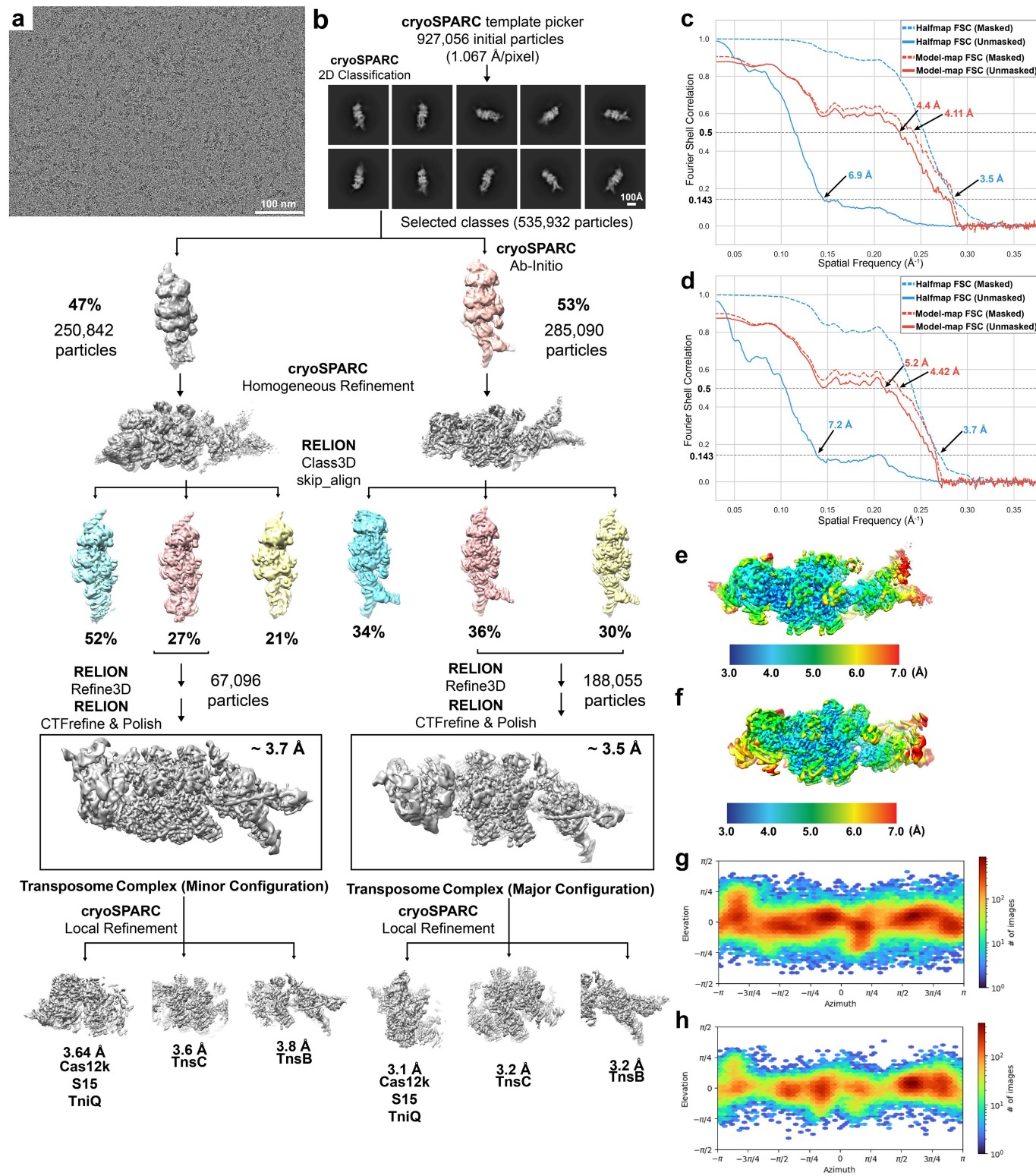

**Extended Data Fig. 2** | See next page for caption.

**Extended Data Fig. 2 | Cryo-EM imaging and image processing pipeline of the transpososome complex. a**. Representative cryo-EM micrograph from the reconstituted transpososome sample. Scale bar (white, bottom right) represents 100 nm. The micrograph shown is an example image from a full dataset consisting of 17,554 micrographs. **b**. Image processing workflow used to analyze the cryo-EM data. 2D classification in cryoSPARC v3.3.1 on template picked particles (from 14,017 micrographs) resulted in 535,932 particles. *Ab-initio* reconstruction on the initial particle stack resulted in two classes, one with 53% of the particles (pink) and the other with 47% of the particles (gray). The two classes were separated for subsequent classification and refinement steps. Before performing 3D classification in RELION v4, each class underwent homogenous refinement in cryoSPARCv3.3.1[35]. RELION v4 3D classification (without alignments, skip_align)[36,37] was applied to both populations from the *ab-initio* reconstruction resulting in the colored volumes shown (blue, pink, and yellow). On the left, the two classes (pink and yellow) that have the best resolved Cas12k density were combined for downstream refinement to produce the final 3D reconstruction (boxed), which is the major configuration of the transpososome complex with 12 TnsC protomers. Local refinement was performed on three different segments of the map, focusing on: Cas12k (3.1 Å), TnsC (3.2 Å) and TnsB (3.2 Å). On the right, the class that has the best resolved Cas12k and TnsB density (27% of particles, shown in pink) was selected for downstream refinement to produce the final 3D reconstruction (boxed), which is the minor configuration of the transpososome complex with 13 TnsC protomers. Similar local refinement was performed on three different segments of the minor TnsC complex to result in high resolution reconstructions of the target site proteins (Cas12k+TniQ+S15), TnsC, and TnsB. **c-d**. Fourier shell correlation (FSC) curve of the major (**c**) and minor (**d**) configuration of the transpososome complex, respectively. Masked (dashed) or unmasked (solid) gold standard half-map FSC (blue) and model-map FSC (red) curves are shown for the refined reconstruction and atomic model. Model-map cutoff (0.5) and gold-standard FSC cutoffs (0.143) are indicated with dashed lines. Estimated resolution based on these cutoffs are indicated. **e-f**. Local resolution filtered reconstruction for the major (**e**) and minor (**f**) configuration of the transpososome complex, respectively, are shown with estimated local resolution indicated using colored surface. Local resolution ranges from 3.0 Å (blue) to 7.0 Å (red). Legend at the bottom indicates local resolution range and values in Angstrom. **g-h**. Angular distribution plot for particle projections of the major (**g**) and minor (**h**) configuration of the transpososome complex, respectively. The plot was calculated in cryoSPARC v3.3.1 and shows the number of particles for each viewing angle. Colors indicate counts; red corresponds to high particle counts for that particular viewing angle, blue to low particle counts.

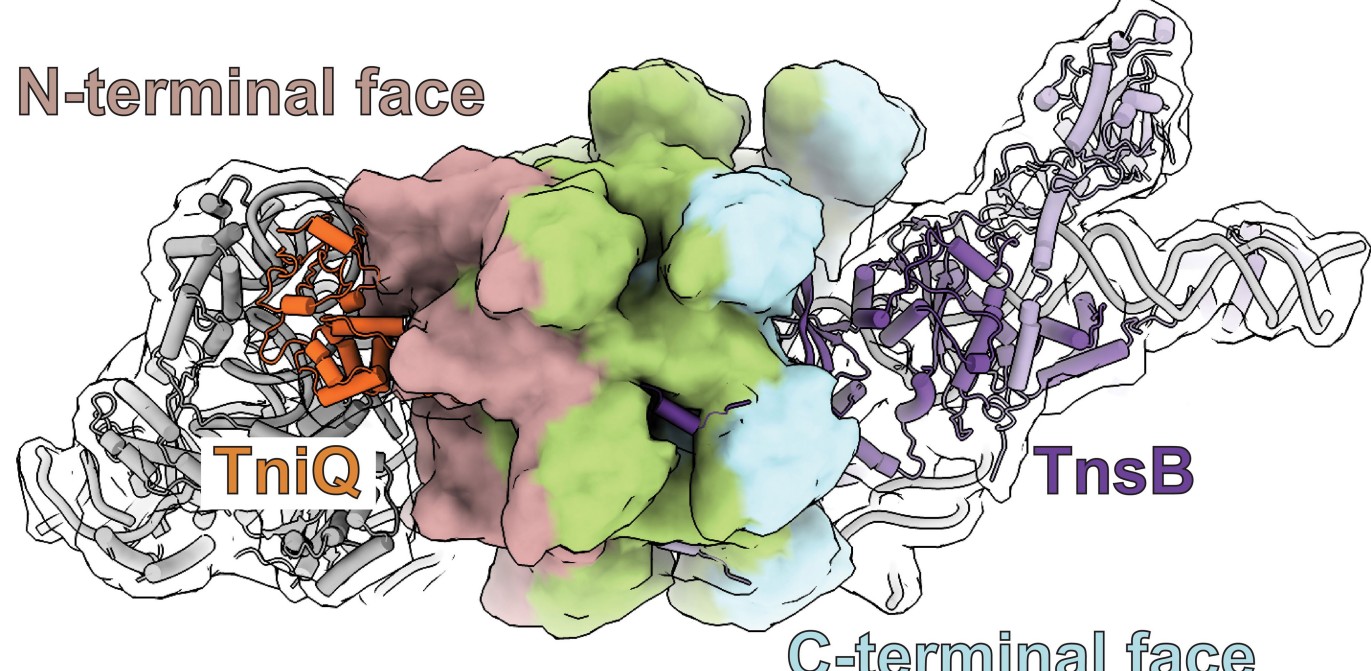

**Extended Data Fig. 3 | TnsC has dedicated faces for TniQ and TnsB within the transpososome.** A simulated map is colored by different regions of TnsC (opaque surface) to represent the N- and C-terminal face of TnsC. The regions are colored as follows: N-terminal face (residues 19–140 on TnsC7-TnsC12, rose-brown), and C-terminal face (residues 141–275 on TnsC1-TnsC6, light blue). Region not included in either N- or C-terminal face is colored green. TniQ (orange ribbon) and TnsB (purple ribbon) associates with the N-terminal face and C-terminal face respectively.

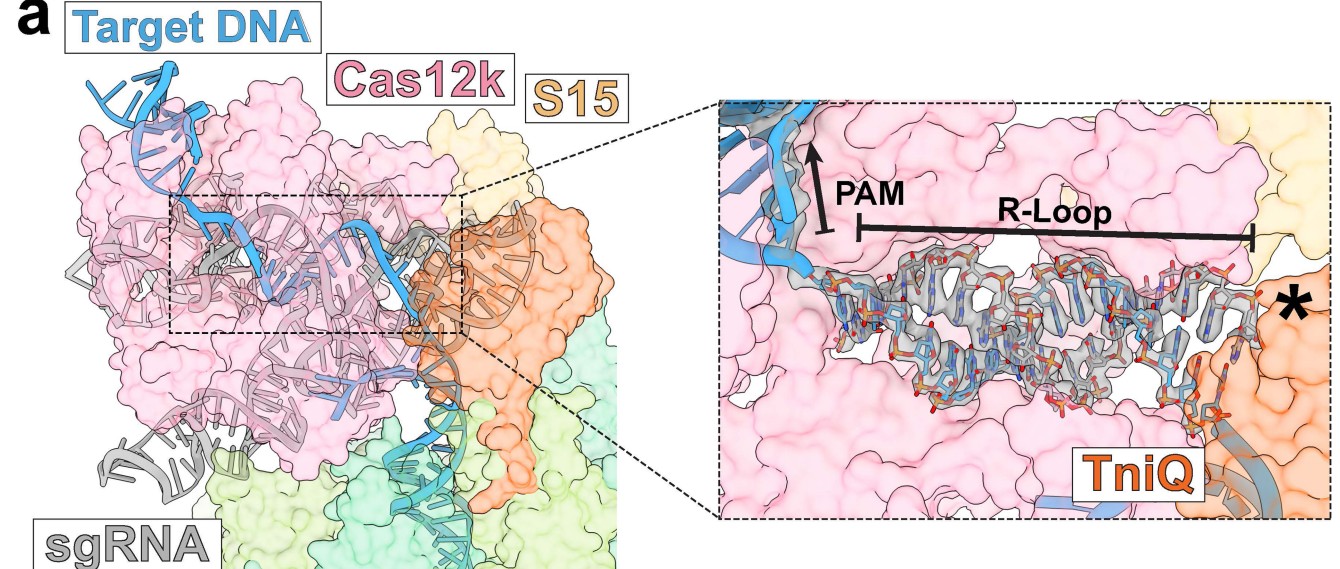

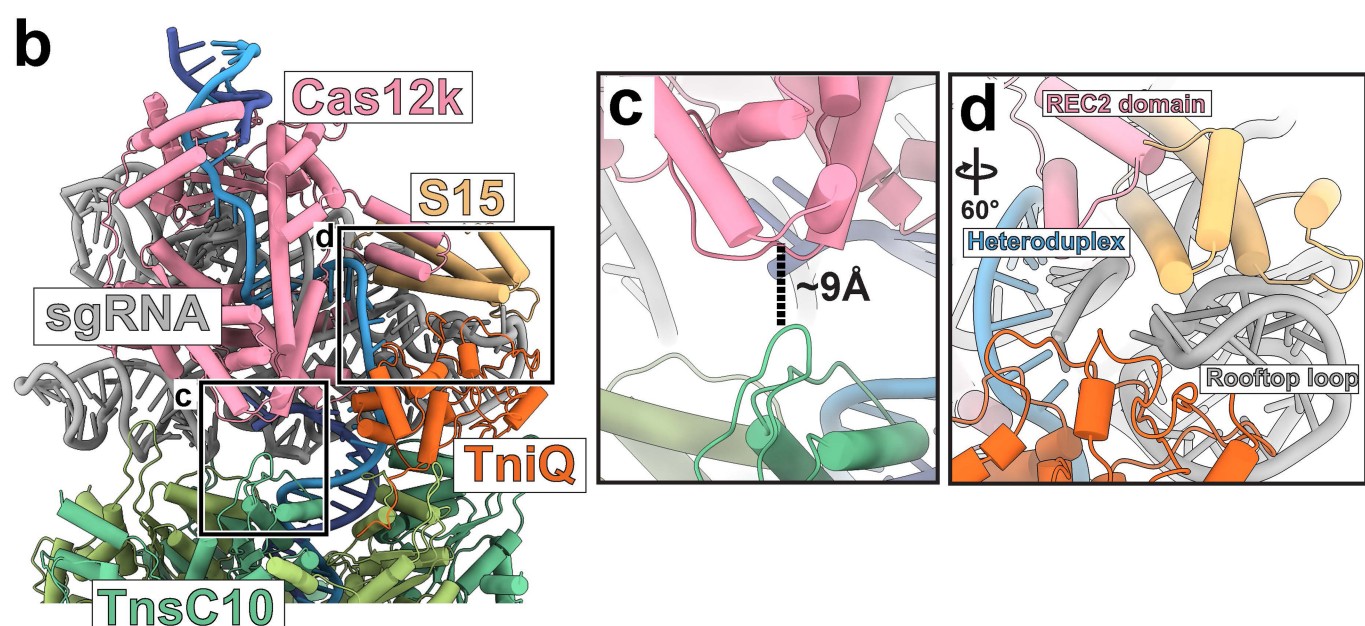

**Extended Data Fig. 4 | Interactions between transpososome components near R-loop. a**. DNA (blue ribbon) forms 17 base pair heteroduplex with RNA (gray) in Cas12k (pink). Cas12k, S15 (tan), and TniQ (orange) are shown in surface representation. Close-up view on the right shows the model docked into the cryo-EM density of the R-loop. The PAM distal end of the R-loop is indicated with an asterisk (*). **b**. Atomic model of the ShCAST transpososome, focusing on Cas12k. The TnsC protomer closest to Cas12k (TnsC10) is labeled. TnsC protomers are numbered as previously defined (see Main Text, Fig. 2). ShCAST

protein and nucleic acid components are labeled and colored according to previously defined colors (see Main Text, Fig. 1). Black box indicates the TnsC finger loop that is close to Cas12k shown as inset in panel B. **c**. The closet distance between TnsC protomer TnsC10 and Cas12k is shown with dashed line and labeled (in Å). **d**. S15 (beige) is positioned between the REC2 domain of Cas12k (pink(and the sgRNA-DNA heteroduplex (blue/grey). The rooftop loop of sgRNA (grey) is stabilized by S15 (beige) and TniQ (orange). Rotation with respect to panel a is indicated in top left corner.

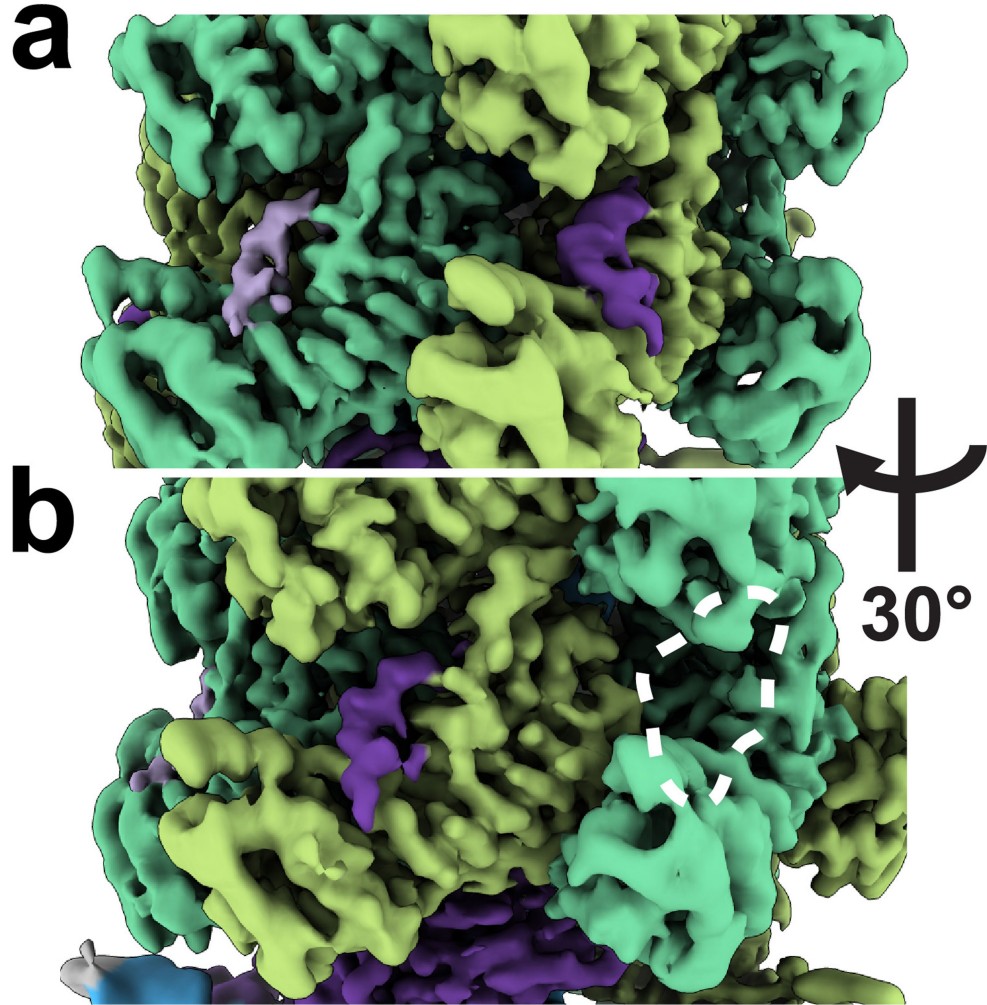

**Extended Data Fig. 5 | TnsB^Hook density occupies select TnsC protomers in the TnsC hexamer closest to TnsB. a**. Local resolution filtered cryo-EM reconstruction (same as that shown in Fig. 1) colored by assignment reveals that TnsB^Hook (light and dark purple) occupies binding sites on TnsC (green). **b**. Rotation of the cryo-EM reconstruction by 30° shows the adjacent TnsB^Hook binding pocket on TnsC (white dashed lines) is empty.

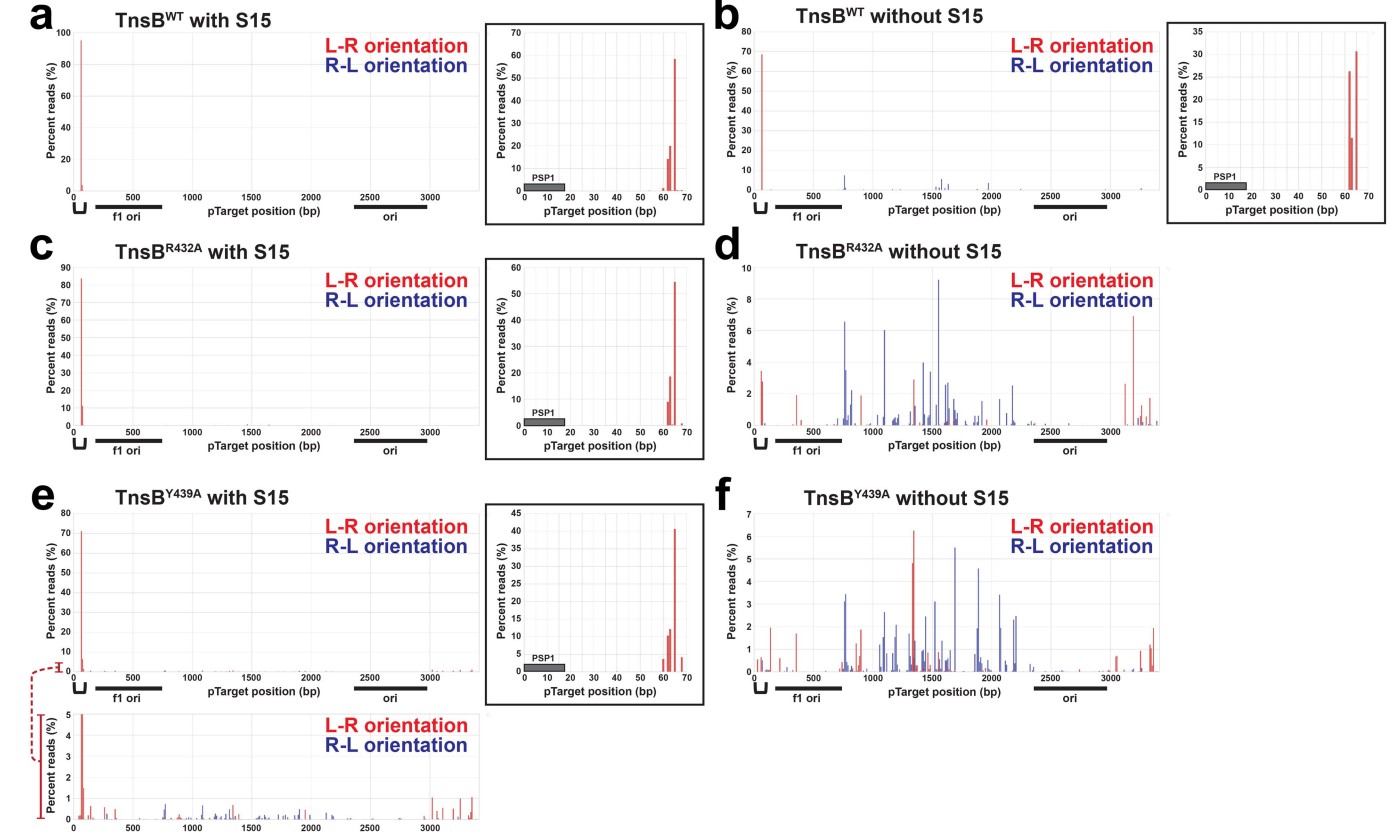

**Extended Data Fig. 6 | High throughput mapping of the *in vitro* transposition events reveals that the identified interaction between TnsC and TnsB IIβ domain is crucial for target-site selection.** Insertion positions were determined by Illumina sequencing of the plasmids extracted from colonies from *in vitro* transpositions under the following conditions (same with Fig. 3f, see Methods): **a**. TnsB wild-type (WT) with S15, **b**. TnsB WT without S15, **c**. TnsB R432A with S15, **d**. TnsB R432A without S15, **e**. TnsB Y439A with S15, and **f**. TnsB Y439A without S15. Determined insertion positions were plotted as a histogram indicating the percentage of the reads at the 10 base-pair (bp) windows within the target plasmid. Position numbers (x-axis) correspond to the number of base pairs between the PAM and the beginning of the transposon-end sequence after the

transposition. Red and blue bars represent the transposition products with the left end-right end (L-R, correct) or the right end-left end (R-L, wrong) orientation respectively. For the conditions with high on-target percentage (> 60%, panels a, b, c, and e), insets are presented for the positions around the PSP1 protospacer (from 0 bp to 70 bp), which is indicated with black brackets on the x-axis. Grey bar in the inset indicates the 17 bp PSP1 protospacer. For panel e, a red bar on the y-axis represents the region for zoom-in on the lower panel to visualize signals from the off-target transposition events. Two origins of replications within the target plasmids (f1 ori and ori) are annotated as black bars on the x-axis, which explains the reason for the cold spots for the transpositions.

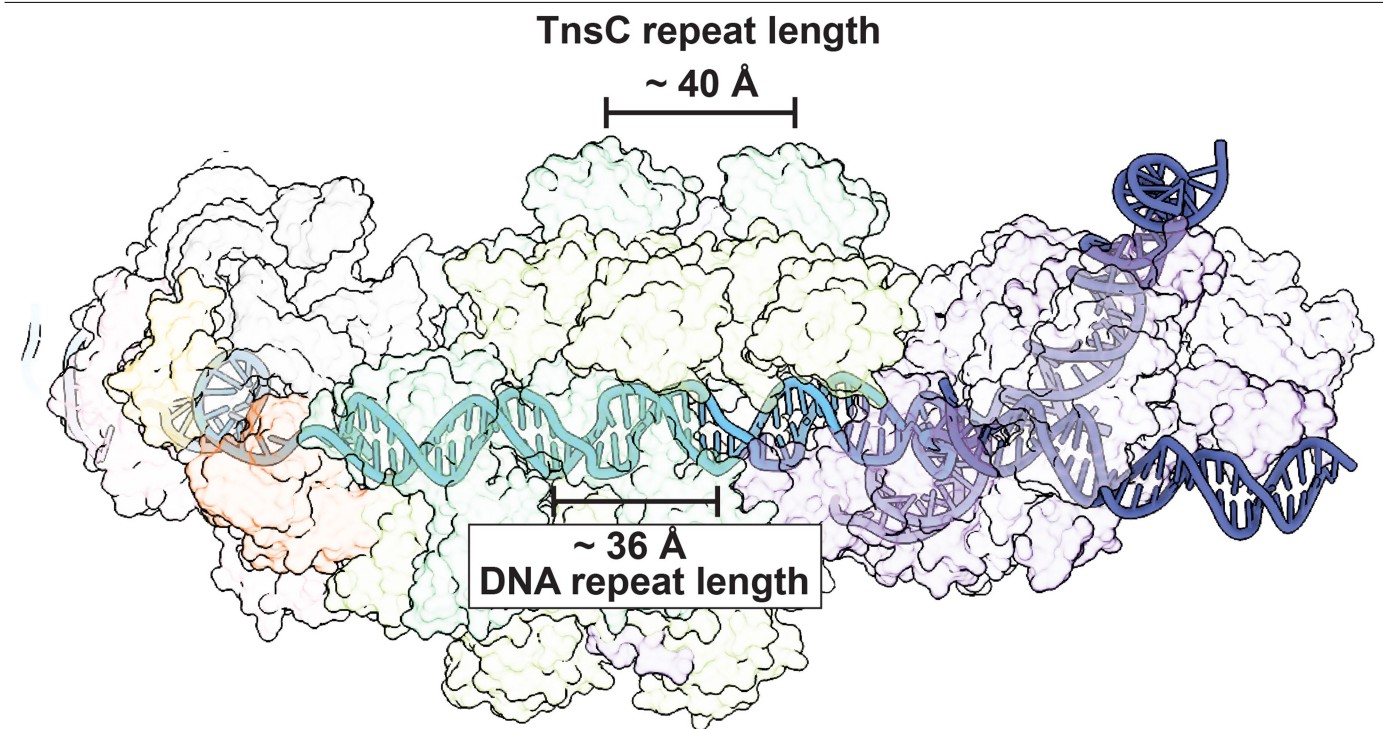

**Extended Data Fig. 7 | TnsC in ShCAST transpososome does not match helical parameters of the bound DNA.** The repeat length of TnsC turn (~6 protomers per turn) is approximately 40 Å, while the repeat length of the TnsC-bound DNA (~11 base pairs per turn) is approximately 36 Å. DNA model is represented in a solid ribbon. Protein components and sgRNA in the transpososome are represented as transparent surfaces. Color scheme is identical to the established colors in Fig. 1.

# a

**TnsC13**

**TnsB**

# b Major configuration

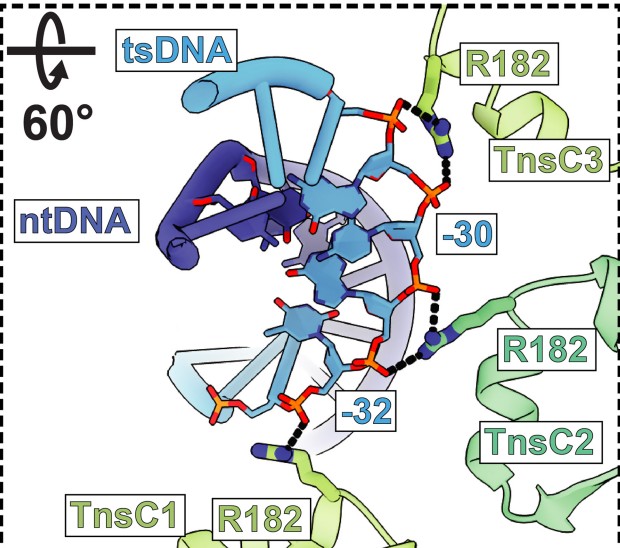

60°

tsDNA

ntDNA

R182

TnsC3

-30

R182

TnsC2

-32

TnsC1    R182

# Minor configuration

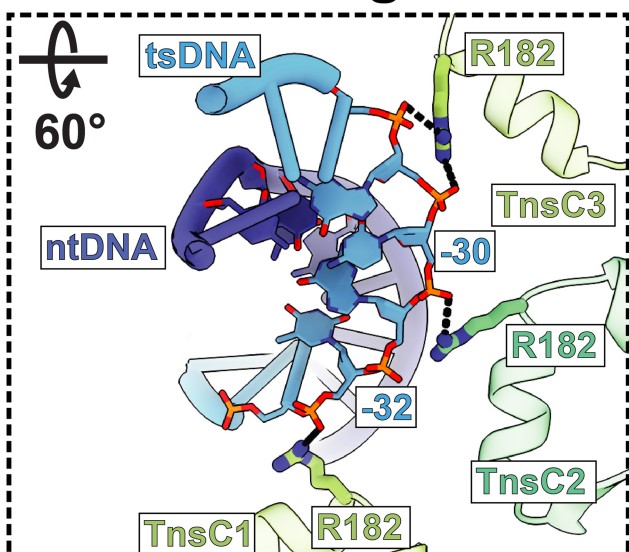

60°

tsDNA

ntDNA

R182

TnsC3

-30

R182

-32    TnsC2

TnsC1    R182

**Extended Data Fig. 8 | TnsC-DNA interaction proximal to TnsB is identical in the major and minor configurations. a**. Low pass filtered (10 Å) cryo-EM reconstructions of both major (transparent grey) and minor (solid, colored) configurations are aligned with respect to TnsB. The dashed box indicates the TnsC-DNA interactions at the TnsB proximal region shown as inset in panel B. **b**. Three TnsB-proximal TnsC protomers (From TnsC1 to TnsC3) in both major and minor configurations interact with target strand DNA (tsDNA) in an identical manner through residue R182. Hydrogen bonding interactions (distance cut off <4 Å) between protein residues and the sugar-phosphate backbone of DNA are represented with dashed lines. Base pairs that are interacting with TnsC are represented as filled nucleotides and stick phosphate-backbone.

**Extended Data Table 1 | Cryo-EM Map and Model Validation**

| Name | TnsB$^{CTD}$-TnsC-TniQ complex | Transpososome major configuration | Transpososome minor configuration |
|---|---|---|---|
| PDB ID | 7SVU | 8EA3 | 8EA4 |
| EMDB ID | EMD-25453 | EMD-27971 | EMD- 27972 |
| **Data collection and Processing** | | | |
| Microscope | Talos-Arctica | Titan-Krios | Titan-Krios |
| Voltage (kV) | 200 | 300 | 300 |
| Camera | K3 | K3 | K3 |
| Magnification | 63,000 | 81,000 | 81,000 |
| Pixel size at detector (Å/pixel) | 1.330 | 1.067 | 1.067 |
| Electron exposure (e$^-$/Å$^2$) | 50.00 | 49.91 | 49.91 |
| Exposure rate (e-/pixel/sec) | 28.0 | 21.9 | 21.9 |
| Number of frames | 50 | 50 | 50 |
| Defocus range (μm) | -1 – -2.5 | -0.8 – -2.5 | -0.8 – -2.5 |
| Automation software | SerialEM | Leginon | Leginon |
| Energy filter slit width | 20 eV | 20 eV | 20 eV |
| | | | |
| **For each reconstruction:** | | | |
| Refined particles (no.) | 624,597 | 285,090 | 250,842 |
| Final particles (no.) | 61,515 | 188,055 | 67,096 |
| Symmetry | C1 | C1 | C1 |
| Resolution (global, Å) | | | |
| FSC 0.5 (unmasked/masked) | 7.4/4.0 | 8.8/3.9 | 9.6/4.2 |
| FSC 0.143 (unmasked/masked) | 4.1/3.5 | 6.9/3.5 | 7.2/3.7 |
| Resolution range (local, Å) | 3.0 – 7.0 | 3.0 – 7.0 | 3.0 – 7.0 |
| Resolution range due to anisotropy (Å) | 3.2 – 3.7 | 3.6 – 4.1 | 3.8 – 4.4 |
| Map sharpening $B$ factor (Å$^2$) | - | 85.3 | 70.7 |
| | | | |
| **Model composition** | | | |
| Protein residues | 3,283 | 5,618 | 5,875 |
| Ligands | 22 | 26 | 28 |
| RNA/DNA | 52 | 495 | 495 |
| | | | |
| **Model Refinement and Validation** | | | |
| Refinement package | Coot/Rosetta/Phenix | Coot/Rosetta/Phenix | Coot/Rosetta/Phenix |
| - resolution cutoff (Å) | 3.5 | 3.2 | 3.8 |
| Model-Map scores | | | |
| - Average FSC (0.5 cutoff, Å) | 3.89 | 4.11 | 4.42 |
| R.m.s deviations from ideal values | | | |
| Bond length (Å) | 1.826 | 1.714 | 1.665 |
| Bond angles (°) | 0.019 | 0.012 | 0.012 |
| MolProbity score | 1.75 | 1.68 | 1.68 |
| CaBLAM outliers (%) | 1.38 | 1.2 | 1.00 |
| Clashscore | 14.46 | 11 | 13 |
| Poor rotamers (%) | 0.42 | 0.22 | 0.21 |
| C-beta outliers (%) | 0.39 | 0.23 | 0.22 |
| EMRinger score | 1.75 | 1.50 | 1.19 |
| Ramachandran plot | | | |
| Favored (%) | 97.59 | 97.34 | 97.47 |
| Allowed (%) | 2.26 | 2.56 | 2.38 |
| Disallowed (%) | 0.15 | 0.18 | 0.15 |

**Extended Data Table 2 | The distance between TnsC residues and the closest atom from the phosphate backbone of DNA**

| TnsC | Lys103 DNA | Lys103 Distance | Thr121 DNA | Thr121 Distance | Lys150 DNA | Lys150 Distance | Arg182 DNA | Arg182 Distance | Lys119* DNA | Lys119* Distance |
|---|---|---|---|---|---|---|---|---|---|---|
| 12 | . | 4.8 Å | nt | 2.2 Å | nt | 3.1 Å | . | 5.7 Å | . | 4.6 Å |
| 11 | ts | 3.9 Å | ts | 3.6 Å | nt | 3.9 Å | . | 5.4 Å | . | 5.1 Å |
| 10 | . | 4.2 Å | . | 4.5 Å | . | 4.5 Å | . | 4.6 Å | . | 6.4 Å* |
| 9 | . | 4.4 Å | . | 5.1 Å | . | 4.7 Å | . | 6.0 Å | . | 4.5 Å* |
| 8 | . | 4.6 Å | . | 5.0 Å | . | 5.1 Å | . | 6.2 Å | . | 4.6 Å* |
| 7 | . | 5.3 Å | . | 5.2 Å | . | 4.9 Å | . | 5.5 Å | . | 4.4 Å* |
| 6 | . | 5.4 Å | . | 5.7 Å | . | 4.6 Å | . | 5.6 Å | ts | 3.1 Å* |
| 5 | . | 4.6 Å | . | 4.3 Å | . | 5.7 Å | . | 4.6 Å | . | 4.3 Å |
| 4 | . | 4.9 Å | nt | 4.0 Å | . | 7.1 Å | . | 5.0 Å | ts | 3.9 Å |
| 3 | . | 4.1 Å | nt | 2.5 Å | . | 5.6 Å | ts | 2.7 Å | ts | 3.1 Å* |
| 2 | . | 4.3 Å | nt | 2.2 Å | . | 4.7 Å | ts | 3.6 Å | ts | 2.6 Å* |
| 1 | . | 5.9 Å | nt | 2.5 Å | . | 4.9 Å | ts | 3.3 Å | . | 4.3 Å |

Distances were measured between potential hydrogen-bonding donor atoms and the closest oxygen atom from the DNA phosphate backbone. Most of the Lys103 residues (11 out of 12) are too far from the DNA backbone to form a hydrogen bonding interaction. Thr121 and Arg182 interact with the non-target strand (nt) or the target-strand (ts) throughout the transpososome. Lys150 contacts the non-target strand near the Cas12k of the transpososome. Lys119 residues are positioned following the target strand, but most of the residues (7 out of 12, indicated with red asterisks) do not have a strong cryo-EM density of the side chain. These residues were built with the most common rotamers for measuring the distance. Black asterisks indicate the uncertainty of the measured distances due to the lack of side-chain density. The closest strand is annotated only when the distance is less than or equal to 4Å, and annotated as a dot (.) otherwise.

# Reporting Summary

## Statistics

For all statistical analyses, confirm that the following items are present in the figure legend, table legend, main text, or Methods section.

| n/a | Confirmed | |
|---|---|---|
| ☐ | ☒ | The exact sample size (*n*) for each experimental group/condition, given as a discrete number and unit of measurement |
| ☐ | ☒ | A statement on whether measurements were taken from distinct samples or whether the same sample was measured repeatedly |
| ☒ | ☐ | The statistical test(s) used AND whether they are one- or two-sided<br>*Only common tests should be described solely by name; describe more complex techniques in the Methods section.* |
| ☒ | ☐ | A description of all covariates tested |
| ☒ | ☐ | A description of any assumptions or corrections, such as tests of normality and adjustment for multiple comparisons |
| ☐ | ☒ | A full description of the statistical parameters including central tendency (e.g. means) or other basic estimates (e.g. regression coefficient) AND variation (e.g. standard deviation) or associated estimates of uncertainty (e.g. confidence intervals) |
| ☒ | ☐ | For null hypothesis testing, the test statistic (e.g. $F$, $t$, $r$) with confidence intervals, effect sizes, degrees of freedom and $P$ value noted<br>*Give P values as exact values whenever suitable.* |
| ☒ | ☐ | For Bayesian analysis, information on the choice of priors and Markov chain Monte Carlo settings |
| ☒ | ☐ | For hierarchical and complex designs, identification of the appropriate level for tests and full reporting of outcomes |
| ☒ | ☐ | Estimates of effect sizes (e.g. Cohen's *d*, Pearson's *r*), indicating how they were calculated |

*Our web collection on statistics for biologists contains articles on many of the points above.*

## Software and code

Policy information about availability of computer code

| | |
|---|---|
| Data collection | SerialEM v4.0, Leginon v3.5, i-control v1.10.4 |
| Data analysis | cryoSPARC v3.3.1, RELION v4, Appion v3.4, Phenix v1.19.1-4122, Coot v0.9.8.2, UCSF Chimera v1.14, UCSF ChimeraX V1.2.5, MolProbity v4.5.1, Warp v1.0.9, BBtools v38.98, DISOPREDE3 |

For manuscripts utilizing custom algorithms or software that are central to the research but not yet described in published literature, software must be made available to editors and reviewers. We strongly encourage code deposition in a community repository (e.g. GitHub). See the Nature Portfolio guidelines for submitting code & software for further information.

## Data

Policy information about availability of data

All manuscripts must include a data availability statement. This statement should provide the following information, where applicable:
- Accession codes, unique identifiers, or web links for publicly available datasets
- A description of any restrictions on data availability
- For clinical datasets or third party data, please ensure that the statement adheres to our policy

Atomic models determined in this study are available through the Protein Data Bank (PDB) with accession codes: 8EA3 (major configuration), 8EA4 (minor configuration), and 7SVU (TnsBCTD-TnsC-TniQ complex). All cryo-EM reconstructions from this study are available through the EMDB with accession codes:

EMD-27971 (major configuration), EMD-27972 (minor configuration), and EMD-25453 (TnsBCTD-TnsC-TniQ complex). Other atomic models used in this study are available through the PDB, including PDB: 7M99 (ATPγS-bound TnsC helical filament), PDB: 7SVW (TnsB strand-transfer complex), and PDB: 7PLA (sgRNA)

## Human research participants

Policy information about studies involving human research participants and Sex and Gender in Research.

| Reporting on sex and gender | N/A |
|---|---|
| Population characteristics | N/A |
| Recruitment | N/A |
| Ethics oversight | N/A |

Note that full information on the approval of the study protocol must also be provided in the manuscript.

## Field-specific reporting

Please select the one below that is the best fit for your research. If you are not sure, read the appropriate sections before making your selection.

☒ Life sciences    ☐ Behavioural & social sciences    ☐ Ecological, evolutionary & environmental sciences

For a reference copy of the document with all sections, see nature.com/documents/nr-reporting-summary-flat.pdf

## Life sciences study design

All studies must disclose on these points even when the disclosure is negative.

| Sample size | Sample sizes are reported in the figure legends. To obtain SD values, biological replicates of n=3 were used. For illumina sequencing of the in vitro transposition products, we pooled all the colonies from the antibiotics-selected plates. The number of colonies ranged from 50 to 500 depending on experimental condition. |
|---|---|
| Data exclusions | No data was excluded. |
| Replication | All the information related to the replication were included in the figure legends. Both ATP-hydrolysis assays and transposition assays were done in biological triplicates, and all the results showed similar results. |
| Randomization | For cryo-EM structures, during the 3D refinement processing, all the particles were randomly split into two half sets. For all other experiments, all the data was used in analysis so randomization was not needed. |
| Blinding | For counting colonies from in vitro transposition assays, each condition was labeled with numbers, and counted by multiple operators without knowing the conditions corresponding to each labeled number. Illumina sequencing was done by operators who do not have knowledge on the system. For ATP hydrolysis assay, each condition was also labeled with numbers, and the experiments and the analysis were done without knowing the condition corresponding to each labels. |

## Behavioural & social sciences study design

All studies must disclose on these points even when the disclosure is negative.

| Study description | N/A |
|---|---|
| Research sample | N/A |
| Sampling strategy | N/A |
| Data collection | N/A |
| Timing | N/A |
| Data exclusions | N/A |
| Non-participation | N/A |

| Randomization | N/A |
|---|---|

# Ecological, evolutionary & environmental sciences study design

All studies must disclose on these points even when the disclosure is negative.

| Study description | N/A |
|---|---|
| Research sample | N/A |
| Sampling strategy | N/A |
| Data collection | N/A |
| Timing and spatial scale | N/A |
| Data exclusions | N/A |
| Reproducibility | N/A |
| Randomization | N/A |
| Blinding | N/A |

Did the study involve field work?  ☐ Yes  ☒ No

# Reporting for specific materials, systems and methods

We require information from authors about some types of materials, experimental systems and methods used in many studies. Here, indicate whether each material, system or method listed is relevant to your study. If you are not sure if a list item applies to your research, read the appropriate section before selecting a response.

## Materials & experimental systems

| n/a | Involved in the study |
|---|---|
| ☒ | ☐ Antibodies |
| ☒ | ☐ Eukaryotic cell lines |
| ☒ | ☐ Palaeontology and archaeology |
| ☒ | ☐ Animals and other organisms |
| ☒ | ☐ Clinical data |
| ☒ | ☐ Dual use research of concern |

## Methods

| n/a | Involved in the study |
|---|---|
| ☒ | ☐ ChIP-seq |
| ☒ | ☐ Flow cytometry |
| ☒ | ☐ MRI-based neuroimaging |

## Antibodies

| Antibodies used | N/A |
|---|---|
| Validation | N/A |

## Eukaryotic cell lines

Policy information about cell lines and Sex and Gender in Research

| Cell line source(s) | N/A |
|---|---|
| Authentication | N/A |
| Mycoplasma contamination | N/A |
| Commonly misidentified lines (See ICLAC register) | N/A |

# Palaeontology and Archaeology

| | |
|---|---|
| Specimen provenance | N/A |
| Specimen deposition | N/A |
| Dating methods | N/A |

☐ Tick this box to confirm that the raw and calibrated dates are available in the paper or in Supplementary Information.

| | |
|---|---|
| Ethics oversight | N/A |

Note that full information on the approval of the study protocol must also be provided in the manuscript.

# Animals and other research organisms

Policy information about studies involving animals; ARRIVE guidelines recommended for reporting animal research, and Sex and Gender in Research

| | |
|---|---|
| Laboratory animals | N/A |
| Wild animals | N/A |
| Reporting on sex | N/A |
| Field-collected samples | N/A |
| Ethics oversight | N/A |

Note that full information on the approval of the study protocol must also be provided in the manuscript.

# Clinical data

Policy information about clinical studies

All manuscripts should comply with the ICMJE guidelines for publication of clinical research and a completed CONSORT checklist must be included with all submissions.

| | |
|---|---|
| Clinical trial registration | N/A |
| Study protocol | N/A |
| Data collection | N/A |
| Outcomes | N/A |

# Dual use research of concern

Policy information about dual use research of concern

## Hazards

Could the accidental, deliberate or reckless misuse of agents or technologies generated in the work, or the application of information presented in the manuscript, pose a threat to:

| No | Yes | |
|---|---|---|
| ☐ | ☒ | Public health |
| ☐ | ☒ | National security |
| ☐ | ☒ | Crops and/or livestock |
| ☐ | ☒ | Ecosystems |
| ☐ | ☒ | Any other significant area |

| | |
|---|---|
| Other impacts | N/A |

| | |
|---|---|
| Hazards | N/A |

For examples of agents subject to oversight, see the United States Government Policy for Institutional Oversight of Life Sciences Dual Use Research of Concern.

## Experiments of concern

Does the work involve any of these experiments of concern:

| No | Yes | |
|----|-----|---|
| ☒ | ☐ | Demonstrate how to render a vaccine ineffective |
| ☒ | ☐ | Confer resistance to therapeutically useful antibiotics or antiviral agents |
| ☒ | ☐ | Enhance the virulence of a pathogen or render a nonpathogen virulent |
| ☒ | ☐ | Increase transmissibility of a pathogen |
| ☒ | ☐ | Alter the host range of a pathogen |
| ☒ | ☐ | Enable evasion of diagnostic/detection modalities |
| ☒ | ☐ | Enable the weaponization of a biological agent or toxin |
| ☒ | ☐ | Any other potentially harmful combination of experiments and agents |

## Precautions and benefits

| | |
|---|---|
| Biosecurity precautions | N/A |
| Biosecurity oversight | N/A |
| Benefits | N/A |
| Communication benefits | N/A |

# ChIP-seq

## Data deposition

☐ Confirm that both raw and final processed data have been deposited in a public database such as GEO.

☐ Confirm that you have deposited or provided access to graph files (e.g. BED files) for the called peaks.

| | |
|---|---|
| Data access links<br>*May remain private before publication.* | *For "Initial submission" or "Revised version" documents, provide reviewer access links. For your "Final submission" document, provide a link to the deposited data.* |
| Files in database submission | *Provide a list of all files available in the database submission.* |
| Genome browser session<br>(e.g. UCSC) | *Provide a link to an anonymized genome browser session for "Initial submission" and "Revised version" documents only, to enable peer review. Write "no longer applicable" for "Final submission" documents.* |

## Methodology

| | |
|---|---|
| Replicates | N/A |
| Sequencing depth | N/A |
| Antibodies | N/A |
| Peak calling parameters | N/A |
| Data quality | N/A |
| Software | N/A |

# Flow Cytometry

## Plots

Confirm that:

☐ The axis labels state the marker and fluorochrome used (e.g. CD4-FITC).

☐ The axis scales are clearly visible. Include numbers along axes only for bottom left plot of group (a 'group' is an analysis of identical markers).

☐ All plots are contour plots with outliers or pseudocolor plots.

☐ A numerical value for number of cells or percentage (with statistics) is provided.

## Methodology

| | |
|---|---|
| Sample preparation | N/A |
| Instrument | N/A |
| Software | N/A |
| Cell population abundance | N/A |
| Gating strategy | N/A |

☐ Tick this box to confirm that a figure exemplifying the gating strategy is provided in the Supplementary Information.

# Magnetic resonance imaging

## Experimental design

| | |
|---|---|
| Design type | N/A |
| Design specifications | N/A |
| Behavioral performance measures | N/A |

## Acquisition

| | |
|---|---|
| Imaging type(s) | N/A |
| Field strength | N/A |
| Sequence & imaging parameters | N/A |
| Area of acquisition | N/A |

Diffusion MRI     ☐ Used     ☒ Not used

## Preprocessing

| | |
|---|---|
| Preprocessing software | N/A |
| Normalization | N/A |
| Normalization template | N/A |
| Noise and artifact removal | N/A |
| Volume censoring | N/A |

## Statistical modeling & inference

| | |
|---|---|
| Model type and settings | N/A |
| Effect(s) tested | N/A |

Specify type of analysis:     ☐ Whole brain     ☐ ROI-based     ☐ Both

| | |
|---|---|
| Statistic type for inference (See Eklund et al. 2016) | N/A |
| Correction | N/A |

## Models & analysis

| n/a | Involved in the study |
|---|---|
| ☒ | ☐ Functional and/or effective connectivity |
| ☒ | ☐ Graph analysis |
| ☒ | ☐ Multivariate modeling or predictive analysis |

