## [Peer Review File · Nature]

Manuscript Title: Structures of the holo CRISPR RNA-guided transposon integration complex

Reviewer Comments & Author Rebuttals

Reviewer Reports on the Initial Version:

Referees' comments:

Referee #1:

This is an exciting work, which provides explanation how a CRISPR-associated transposon is targeted to a precise location. It was known that TnsC bridges the transposase (TnsB) to the targeting assembly (Cas12). Bafflingly, TnsC tends to form long polymers on DNA, which are heterogenous in length. Moreover, TnsB stimulates disassembly of these polymers. Thus, TnsC does not seem to fit the purpose of a molecular ruler that spaces TnsB. The structures of the complete transposition machinery presented in this work explain how a TnsC polymer (anchored on one side to CRISPR targeting assembly) and TnsB cooperate to create a such ruler. Interestingly, there is no single protein chain that spans the distance between TnsB and Cas12. Instead, it is the target DNA what transmits an allosteric signal, which limits disassembly of the TnsC polymer. This a beautiful idea, supported by structural data and biochemistry.

I only have several relatively minor comments/points for discussion:

1. The transposon inserts at a defined distance from a target site, with a specific polarity. This polarity is reproduced in the assays shown in Fig. S11 (red bars). The core transpososome (comprised of TnsB tetramer on donor DNA ends) is a C2-symmetric assembly. Presumably, some asymmetry is introduced by differences in DNA sequences of the left and right transposon ends (?). However, the TnsB-TnsC contacts are primarily via flexible TnsB tails. Do these or prior structures explain the observed polarity of transposition? Maybe this is a trivial point to an expert in the field, but would help to add discussion the origin of insertion polarity.
2. The structure explains recruitment of TnsB to a near precise position. However, just recruitment alone cannot explain the striking specify of transposition. There has to be a mechanism preventing off-site activity of TnsB. Do the current or prior structures shed lights on how such specificity is achieved?
3. "target DNA does not track with TnsC protomers" in the abstract sounds vague. One needs to carefully read the paper to understand what it means.
4. Page 6, Line 143 "RNase H transposase" is vague and imprecise. TnsB belongs to the family of DDE/D transposases, which share the structural fold with RNase H, and is highly similar to MuA transposase.

Referee #2:

CRISPR-associated transposons (CAST) are mobile genetic elements that have acquired CRISPR-like targeting systems to direct their insertions to desired DNA locations in bacteria. They provide a beautiful example of the constant dynamic arms-race between mobile genetic elements and their host, which accompanies the evolution of microbial genomes. In addition, CAST elements offer uniquely promising candidates for novel genome-editing applications, because they allow direct RNA-guided integration of a DNA cargo near a programmable target sequence.

In the current manuscript, Dr. Kellogg and colleagues investigate the molecular machinery of a CAST transposon from the Type V-K group (shCAST). Using cryogenic electron microscopy (cryo-EM), the authors visualise the entire CAST transposition complex (transpososome) in a post-integration state, revealing key features of target site selection, transposase recruitment, and DNA insertion. Together with functional experiments, the results shed first light onto the intricate coordination between target choice and transposon integration, opening unmatched opportunities for the rational design of an effective one-off gene insertion toolset for research and medical applications. The data presented in this study is a key contribution in transposon and CRISPR biology, with broad relevance to diverse areas of mechanistic biology and an undisputable impact in genome engineering technology. In my opinion, this work is likely to form a long-standing highlight in the field and clearly merits publication in Nature.

The structure is high quality and the functional experiments have been well designed and executed. The manuscript is overall well written, but some descriptions are vague (see below); more specific wording would help clarify the deductions, their conclusions and significance. Additionally, I have a couple of questions regarding data presentation, which the authors should consider to address.

1. The Cas part of the complex seems poorly resolved in the EM reconstruction in Fig. 1c. Why is that? Is part of the complex (e.g. part of the RNA) missing or disordered? Please explain. If map quality in Fig. 1c is compromised by flexibility, local refined and/or composite maps may be shown to help appreciate data quality. In fact, the Cas part looks better in Fig. 2a (composite map), but still seems to miss much of the RNA when compared to ref. 10.
2. The binding of S15 and the architecture of the TniQ-Cas connection in the structure should be described briefly. Although these parts of the structure are not the focus of this manuscript, it would help to summarize the principles for completeness.
3. A key question for the mechanism of Tn7 transposition is how TnsB is activated, when it binds to TnsC at the RNA-selected target site. This is also an important question for genetic applications, because on-target activation prohibits off-target integration in the genome. Does the transpososome structure, in comparison with TnsC-unbound TnsB structures or with inactive structures of other transposases, provide clues for the mechanism of TnsB activation?
4. It makes sense that TnsC is not in an ATPase active state in the transpososome, because no TnsC disassembly is wanted at this point. Can you speculate how the TnsB-TnsC interaction would look in the ATPase active state? What prevents the ATPase active state to form in the transpososome as

compared to longer TnsC filaments? Is allosteric communication between the N- and C- faces of the two-turn helix involved?

5. The conformational differences between the TnsC subunits and their deviation from the helical TnsC structure are difficult to follow. Why does disassembly stop at two full turns? Is this arrangement less prone for ATPase activation (see previous point)? Or does this assembly uniquely allow binding of four TnsB molecules in the active configuration? Clarifying these points will help to understand the specific constraints on integration site selection and promote rational CAST design.

7. I am confused about the comparison between the transpososome structures with 12 and 13 TnsC subunits. Why are the structures aligned at TnsB? Why not at Cas12? Considering that the Cas12-gRNA binding occurs at a defined DNA sequence, while the integration site can vary, The position of Cas12 should be more fixed. Am I overlooking something?

8. In the Discussion, the authors compare the TnsC modules of different Tn7-like elements. I wonder, if the size of the TnsC unit in different elements correlates with the spacing between target recognition and integration sites?

Small edits:

- Page 2, line 32: “domestication” should be replaced with “acquisition”.

- Page 2, line 35: Specify “narrow window”.

- The protein-DNA complex assembly is clever. Please cite also the PFV STC structure paper (Maertens et al., Nature, 2010), where the this DNA design was first used.

- Fig. 2d: Briefly describe the assay and define “# of transformants” in the legend. Should the S15 row below the graph show “-+ -+ -+ -+”? So, one experiment without and one with S15 for each TniQ variant?

- Page 6, line 144: In addition to the authors’ own publication, the TnsB STC structure papers (from the Montoya and Nowotny labs), which appeared at about the same time, should also be cited.

- Page 12, lines 298-300: I am not sure how the authors come to the conclusion about the role of TnsC R182 in filament disassembly. Please clarify.

Referee #3:

We already have a detailed view of Tn7 transposition. Meanwhile, the related CAST systems have been painted with broad brush strokes. This manuscript presents important new detail of a CAST system, which advances beyond isolated subunit structures and blob diagrams illustrating their interactions.

The data in Fig. 3 (and lines 174-200) provide insight into the TnsC–TnsC interaction. My only criticism is that the failure of a fragment to stimulate ATPase activity could have many explanations.

The deviation of TnsC from a strict helical structure (lines 210–226) seems like a small point, and the significance was unclear to me, except in so far as it was different from a previous structure. The significance became clearer in the following section, and I wondered why the sections were separate. Since I have a background in protein biochemistry, I tend to worry about the existence of unseen intermediates and the significance of structural snapshots. Notwithstanding, I do appreciate the insights provided by the present structure.

The final section of Results starting on line 268 provides a satisfying explanation for the variable insertion profile.

The model in Fig. 6 is beautiful and represents a significant advance. However, the DNA helix, although stylized, is left-handed. The insights are developed and contextualized in an interesting discussion section.

In the transposition literature the word ‘transpososome’ generally indicates a complex between the transposase and two transposon ends. Here it is defined in the abstract as an integration complex, which is more restrictive than how it is generally understood.

Line items: Many of the following points would be picked up by a sub-editor at Nature. However, the manuscript is slightly outside my field, and I thought it worthwhile to suggest improvements that would help a general reader, such as myself.

Line 17 and elsewhere: THE AAA+ regulator: it is distracting if you drop the definite article.

Lines 17-18: Orientation with respect to what? What is meant by the length of TnsC? Is it a length in Angstroms? Polarity with respect to what?

Line 19, “THE transposase...”: Again, the missing word makes me unsure of the meaning, and I have to read the sentence a couple of times more to make sure I have not misunderstood. In this particular sentence, the problem is compounded by omitting THAT from the phrase “interactions we observe”. Already, I’m finding this manuscript hard going. How can you know that the observed EM interactions stimulate the ATPase activity? Could it not be owing to an unseen intermediate?

Line 21: I could understand how a protein might ‘track’ along DNA, but I’m at a loss to know the meaning of “DNA does not track WITH TnsC protomers.” From the context, I suspect that the authors are trying to say that a previously characterized TnsC homolog tracked round the DNA helix but that the new structure is different.

All in all, I don’t think this abstract is accessible to the general reader, and I include myself in that category.

Line 32: In the transposition field ‘domestication’ is generally used to mean that a component has

become a bona fide member of the community of host genes, where it performs a function and remains under purifying selection.

Line 37: An alternative definition of the 'transpososome' to that offered in Line 14.

Line 39: What is the meaning of the slash in TniQ/TnsD? If it was tniQ-TnsD, I think most readers would take it as a heterodimer.

Lines 38, 42: More missing definite articles. I won't mention this again.

Line 43: "in some cases WAS shown to require". An experimental result should be stated in the past tense. Was shown because the act of showing was done in the past.

Line 44: The quotes round 'matchmaker' are an admission that it is slang. Consider using "bridge."

Line 55, 'obtained a high resolution structure': Was it not stated previously that there was more than one structural conformation resolved? It is best not to mention the minor structure until the point in the results where it is addressed, i.e. line 93.

Line 58: The fact that it is orientation-specific seems hardly worth mentioning. This was established in the 2019 CAST in vitro reconstitution. Even the dCas9-mariner and dCas9-casposon papers from 2019 and 2021 were orientation-specific. It could hardly be otherwise.

The last paragraph of the intro is a better abstract than the abstract, a fact I always tell my students. Writing a good abstract is hard work.

From the start of the Results section, the manuscript becomes an easier read.

First paragraph of the results: if you have space consider citing, or at least mentioning, the development of the integration complex mimic approach.

It might be worth devoting a sentence or two, here or in the discussion, about the integration-complex mimic-approach used here. Transpososomes tend to become increasingly stable as the reaction progresses through the intermediate stages. The approach goes back to the disintegration assays for HIV and V(D)J recombination (see 10.1093/emboj/cdf425 for references and discussion of conformational dynamics).

Line 107: TnsC DOES not FORM. This is followed by another missing word that makes the sentence distracting.

Line 111: Change to "PROBABLY stabilized". Also, consider speculating on the mechanism of stabilization. Does the structure suggest a mechanism? Be careful to avoid tautological reasoning.

Line 128: The word 'interaction' is repeated three times in this sentence.

Line 130, “reported importance”: Is there some uncertainty here? Do you mean ‘putative importance’?

Line 131: Either add supplemental data regarding S15 activity or state it as “data not shown”.

Line 143, “RNaseH transposase”: Will the general reader understand that it has an RNaseH-like fold?

Line 145: Is it necessary to always refer to TnsC as “the AAA+ regulation”? It seems repetitious and the significance of the emphasis is unclear.

Line 149: How predicted? One would expect a flexible linker to be disordered because it would not otherwise be flexible. Sorry, I’m just saying that it’s hardly worth mentioning.

Lines 149–152: The hook seems to be defined more than once.

Line 164, “TnsC’s”: Here and elsewhere, it is best to avoid using the possessive form of an inanimate object. Just write “stimulate the ATPase activity of TnsC”. This even saves you five characters!

Line 167: Excitingly? Significantly might be better. Less is more.

Line 168, “we observe”: I think “there is” is more impactful.

Line 172: Another set of quotes acknowledging the use of slang. Transduce?

Lines 175-181: This section contains six repetitions of “ATP hydrolysis activity.” You’ll save a lot of characters by replacing it by ATPase and even more by eliminating some of the repetition. At about this point I began to become irritated by the use of the term “construct”, which is lab slang.

Line 179, “mutant TnsB insertions”: Is mutant TnsB being inserted into something?

Line 245: Citation needed for “previously identified DNA binding residues.

Author Rebuttals to Initial Comments:

Dear Editor,

We thank the reviewers for their thoughtful and helpful comments. Incorporating revisions based on their feedback has substantially improved the quality and clarity of the manuscript.

We summarize the reviewer feedback as follows:

1. Generally, the main text needs clarification throughout. For example, architecture is not completely described and structural changes with respect to previous structures are not clearly articulated.
2. Terms with slightly different meaning are used interchangeably throughout.
3. The significance of the transpososome TnsC deviating from helical TnsC is not clear.
4. The implications of this study and open questions need to be clearly stated.
5. The abstract is not clearly written and does not present the material in a way that is understandable by the general audience.

We have rewritten the abstract and main text to generally address these issues. We also provide a point-to-point response below.

Referee #1:

This is an exciting work, which provides explanation how a CRISPR-associated transposon is targeted to a precise location. It was known that TnsC bridges the transposase (TnsB) to the targeting assembly (Cas12). Bafflingly, TnsC tends to form long polymers on DNA, which are heterogenous in length. Moreover, TnsB stimulates disassembly of these polymers. Thus, TnsC does not seem to fit the purpose of a molecular ruler that spaces TnsB. The structures of the complete transposition machinery presented in this work explain how a TnsC polymer (anchored on one side to CRISPR targeting assembly) and TnsB cooperate to create such ruler. Interestingly, there is no single protein chain that spans the distance between TnsB and Cas12. Instead, it is the target DNA what transmits an allosteric signal, which limits disassembly of the TnsC polymer. This a beautiful idea, supported by structural data and biochemistry.

We thank Reviewer 1 for his/her appreciation of the work presented here.

I only have several relatively minor comments/points for discussion:

1. The transposon inserts at a defined distance from a target site, with a specific polarity. This polarity is reproduced in the assays shown in Fig. S11 (red bars). The core transpososome (comprised of TnsB tetramer on donor DNA ends) is a C2-symmetric assembly. Presumably, some asymmetry is introduced by differences in DNA sequences of the left and right transposon ends (?). However, the TnsB-TnsC contacts are primarily via flexible TnsB tails. Do these or prior structures explain the observed polarity of transposition? Maybe this is a trivial point to an expert in the field, but would help to add discussion the origin of insertion polarity.

We find this a fascinating open question. The strict preference for L-R insertion polarity (shown in Extended Data Figure 8) indicates that there likely is asymmetry in how the synaptic complex is recruited to the target-site. However, this and all previous integration complex structures (Park et al. PNAS 2022, Tenjo-Castaño F et al. Nat Comms. 2022) consist of a symmetric DNA substrate containing the first two TnsB binding sites, which the reviewer correctly points out is expected to be C2-symmetric. Therefore, this open question is not answered by existing structures and requires further investigation. Unfortunately, due to length restrictions we are unable to expand the discussion to comment on this open question in extensive detail.

We have added clarification expanding on our substrate design to further emphasize the relationship between the substrate used here and in previous studies (Page 4, Lines 83 – 89):

“Our designed DNA substrate contains transposon DNA up to the first two internal TnsB binding sites from the right and left ends(Extended Data Fig. 1b), identical to previous studies^{17,22}. The first 30 base pairs have identical TnsB binding sites on either end¹⁷, in contrast to the subsequent internal TnsB binding sites which are irregularly spaced in right compared to left transposon ends. Therefore, the transposon sequences we have included in our designed substrate most likely do not contribute to the remarkable ability of ShCAST to discriminate insertion orientation². ”

2. The structure explains recruitment of TnsB to a near precise position. However, just recruitment alone cannot explain the striking specificity of transposition. There has to be a mechanism preventing off-site activity of TnsB. Do the current or prior structures shed lights on how such specificity is achieved?

Our previous studies, which focused on TnsC (Park et al., Science 2021), included an investigation of the role of nucleotide in promoting target-site selection in the ShCAST

element. Strikingly, we discovered that, TnsC and TnsB in the presence of ATP exhibited non-targeted transposition activity at levels 5-fold lower than that observed with all components in the presence of ATP (shown in Figure 1B of Park et al. Science 2021). Taken together with the striking similarities between ShCAST and bacteriophage Mu, we believe that the basis of off-site targeting is TnsB and TnsC acting independent of Cas12k. One complication in this line of reasoning is that we observe that the addition of S15 (or possibly other factors, since we did not exhaustively test all possibilities) generally promotes on-site targeting (Compare WT conditions in Figure 3f, on-site transposition ratio was 99% vs 69% with or without S15, respectively) which suggests that stabilization of the R-loop also plays a significant role in promoting transposition at the target site. Therefore, while we have some ideas about the basis of off site targeting, we believe the mechanism of off site targeting is likely to be multi-faceted and possibly involve one or more mechanisms. While we would love to comment on this more definitively, this question will require focused investigation which is outside the scope of this study.

3. “target DNA does not track with TnsC protomers” in the abstract sounds vague. One needs to carefully read the paper to understand what it means.

We have rewritten the abstract to be clearer.

4. Page 6, Line 143 “RNase H transposase” is vague and imprecise. TnsB belongs to the family of DDE/D transposases, which share the structural fold with RNase H, and is highly similar to MuA transposase.

We have revised the sentence to be more precise (Page 7, Line 154):

“TnsB belongs to the family of DDE/D transposase, and bears significant similarities to MuA from bacteriophage Mu”

Referee #2:

CRISPR-associated transposons (CAST) are mobile genetic elements that have acquired CRISPR-like targeting systems to direct their insertions to desired DNA locations in bacteria. They provide a beautiful example of the constant dynamic arms-race between mobile genetic elements and their host, which accompanies the evolution of microbial genomes. In addition, CAST elements offer uniquely promising candidates for novel genome-editing applications, because they allow direct RNA-guided integration of a DNA cargo near a programmable target sequence.

In the current manuscript, Dr. Kellogg and colleagues investigate the molecular machinery of a CAST transposon from the Type V-K group (shCAST). Using cryogenic electron microscopy (cryo-EM), the authors visualise the entire CAST transposition complex (transpososome) in a post-integration state, revealing key features of target site selection, transposase recruitment, and DNA insertion. Together with functional experiments, the results shed first light onto the intricate coordination between target choice and transposon integration, opening unmatched opportunities for the rational design of an effective one-off gene insertion toolset for research and medical applications. The data presented in this study is a key contribution in transposon and CRISPR biology, with broad relevance to diverse areas of mechanistic biology and an undisputable impact in genome engineering technology. In my opinion, this work is likely to form a long-standing highlight in the field and clearly merits publication in Nature. The structure is high quality and the functional experiments have been well designed and executed.

We thank Reviewer 2 for his/her kind words in appreciation of this work.

The manuscript is overall well written, but some descriptions are vague (see below); more specific wording would help clarify the deductions, their conclusions and significance. Additionally, I have a couple of questions regarding data presentation, which the authors should consider to address.

1. The Cas part of the complex seems poorly resolved in the EM reconstruction in Fig. 1c. Why is that? Is part of the complex (e.g. part of the RNA) missing or disordered? Please explain. If map quality in Fig. 1c is compromised by flexibility, local refined and/or composite maps may be shown to help appreciate data quality. In fact, the Cas part looks better in Fig. 2a (composite map), but still seems to miss much of the RNA when compared to ref. 10.

This complex is incredibly large, reaching ~ 1MDa in size. Therefore, it is unsurprising that slight flexibility along the target DNA would amplify alignment errors (see Extended Data Figure 3). This causes what reviewer 2 points out as poorly resolved density in the Cas12k region of the map. In contrast to reviewer 2's view, we find it quite significant that all components can be unambiguously resolved in the overall map. This speaks to the remarkable homogeneity and stability of the transpososome complex, especially compared to the heterogeneity of previous assemblies: TniQ-TnsC (Park et al., *Science* 2021) and Cas12k-TniQ-TnsC (Schmitz et al., *bioRxiv* 2022). Because of this, it is important to show the overall map without any specialized refinement procedures in Figure 1.

Along these lines, we agree with reviewer 2's point and have used local refinement to improve the appearance of the Cas12k end of the complex. Focused refinement on Cas12k significantly improves the resolution as shown in Figure 2. RNA density appears incomplete because of the view we have chosen to show, however the map resulting from focused refinement is consistent with previous Cas12k structures (7PLA and 7N3P). To demonstrate this point, we have added a description of the focused refinement in the main text (Page 5, line 124), and additional panels in Supplementary Figure 1a-b to highlight the completeness of the Cas12k-associated RNA density, which we estimate is 3.1 Å based on gold-standard FSC calculations.

We have clarified the main text as follows (Page 5, Line 124-127):

"Due to slight flexibility in the DNA substrate, the distal ends of the transpososome have lower local resolution (5-7 Å, Extended Data Figure 3e). As expected, local refinement focused on the Cas12k proximal region significantly improved the quality of the reconstruction (Supplementary Fig. 1a-b)."

2. The binding of S15 and the architecture of the TniQ-Cas connection in the structure should be described briefly. Although these parts of the structure are not the focus of this manuscript, it would help to summarize the principles for completeness.

We agree that a summary of S15 and connections between TniQ and Cas are important for completeness. We have added the following description (Page 6, Lines 129 – 134):

"TniQ primarily interacts with TnsC and RNA, consistent with the productive recruitment complex²². S15 is nestled between the REC2 domain of Cas12k and the PAM-distal sgRNA-DNA heteroduplex (Extended Data Fig. 6c). The rooftop loop of the sgRNA is flanked on either side by S15 and TniQ, respectively. TniQ bridges the two TnsC protomers closest to Cas12k (Fig. 2a & Supplementary Fig. 2), however the TniQ-

TnsC12 interface is much smaller than the TniQ-TnsC11 interface (325 Å² vs 915 Å², see Methods). "

3. A key question for the mechanism of Tn7 transposition is how TnsB is activated, when it binds to TnsC at the RNA-selected target site. This is also an important question for genetic applications, because on-target activation prohibits off-target integration in the genome. Does the transpososome structure, in comparison with TnsC-unbound TnsB structures or with inactive structures of other transposases, provide clues for the mechanism of TnsB activation?

This is an important question that was also brought up by reviewer 1. Please see the response to point 2 from reviewer 1.

4. It makes sense that TnsC is not in an ATPase active state in the transpososome, because no TnsC disassembly is wanted at this point. Can you speculate how the TnsB-TnsC interaction would look in the ATPase active state? What prevents the ATPase active state to form in the transpososome as compared to longer TnsC filaments? Is allosteric communication between the N- and C- faces of the two-turn helix involved?

We find this a fascinating question. While the ATPase activity assay results we've included hint that domain II β is likely involved in stimulating ATP hydrolysis activity, the state we have captured is not capable of further stimulating ATPase activity. Therefore, we have too little information at this point to accurately speculate on what the active form looks like. We have added a sentence in the discussion outlining the exciting open questions brought up by the reviewers (Page 13, Line 312-3):

"... the detailed mechanism of how TnsC activates ATPase activity, resists filament disassembly, and stimulates integration requires further investigation."

5. The conformational differences between the TnsC subunits and their deviation from the helical TnsC structure are difficult to follow.

We have rewritten the main text to be clearer with respect to the structural changes, see section titled: "Transpososome TnsC-DNA interactions" (Page 9, Lines 222 – 232).

Why does disassembly stop at two full turns? Is this arrangement less prone for ATPase activation (see previous point)? Or does this assembly uniquely allow binding of four TnsB molecules in the active configuration? Clarifying these points will help to

understand the specific constraints on integration site selection and promote rational CAST design.

Based on our negative stain imaging of the target pot, which represents the state of the reconstitution prior to the addition of donor-bound TnsB (Extended Data Figure 2), we observe a heterogeneous population containing TnsC filaments of variable length far exceeding the two turns we observe in the transpososome. Therefore, ATP hydrolysis must have occurred to create the uniform assemblies we see here. We have clarified this as follows (Page 5, Lines 106 – 111):

"The defined oligomeric assembly of TnsC is significant, because the ShCAST recruitment complex (containing all components except TnsB) consists of heterogeneous assemblies of TnsC, distinguished by the direction of TnsC filaments bound to DNA¹⁹ and the number of turns of TnsC. In contrast to the heterogeneity of the recruitment complex (captured in the target pot reconstitution, Extended Data Fig. 2a), transpososome particles reveal uniform TnsC directionality..."

The two full turns we observe are the result of the substrate we designed, since activation of TnsB for integration shuts off ATPase activity through a mechanism that remains unresolved. Because we have only included four TnsB binding sites, we are unable to rule out whether this assembly uniquely accommodates a defined number of TnsB. We have added additional information regarding the substrate designed to clarify this point (Page 4, Lines 84 – 86):

"Our designed DNA substrate contains transposon DNA up to the first two internal TnsB binding sites from the right and left ends (Extended Data Fig. 1b)."

7. I am confused about the comparison between the transpososome structures with 12 and 13 TnsC subunits. Why are the structures aligned at TnsB? Why not at Cas12? Considering that the Cas12-gRNA binding occurs at a defined DNA sequence, while the integration site can vary, The position of Cas12 should be more fixed. Am I overlooking something?

Surprisingly, the local structural differences we observe in the two transpososome structures are most dramatic close to Cas12k. In contrast, the local structure in the vicinity of TnsB is virtually identical in both structures. We chose to align on TnsB to emphasize these observations and to make comparisons clear. We have reorganized the main text to highlight our reasoning (Page 11, Lines 281 – 285):

"Comparing the two configurations, the specific TnsC-DNA interactions in protomers adjacent to TnsB are virtually identical since the same interactions are made at the same positions on the target DNA substrate (Extended Data Fig. 10). Aligning both transpososome reconstructions (major and minor configurations) based on TnsB reveals an additional TnsC protomer (TnsC13) next to Cas12k..."

8. In the Discussion, the authors compare the TnsC modules of different Tn7-like elements. I wonder, if the size of the TnsC unit in different elements correlates with the spacing between target recognition and integration sites?

Tn7 TnsC forms a heptamer in isolation, not a filament (Shen et al, *NSMB* 2022). The results presented here clearly demonstrate that the behavior of TnsC in isolation is insufficient to account for behavior of TnsC in the context of the transpososome. Therefore, we have insufficient information to accurately predict the relationship between TnsC homologs and spacing. However, we do note that this question brings up very exciting new questions that represent new avenues of research for the field to answer.

Small edits:

- Page 2, line 32: "domestication" should be replaced with "acquisition".

We have revised the text according to the reviewer's suggestion.

- Page 2, line 35: Specify "narrow window".

We have additional information to clarify the narrow window (Page 2, Line 38):

"DNA cargo is inserted in a single orientation, with defined spacing from the protospacer adjacent motif (PAM), and within a narrow window (5 – 10 bp)"

- The protein-DNA complex assembly is clever. Please cite also the PFV STC structure paper (Maertens et al., *Nature*, 2010), where this DNA design was first used.

We have included this reference as well as a similar study (Yin and Craigie, *Protein Science*, 2012) and highlighted this in the main text (Page 4, Lines 79 – 81):

"This principle was successfully employed in the past to stabilize strand-transfer complex structures of other integrases/transposases, such as PFV^{20,21}."

- Fig. 2d: Briefly describe the assay and define “# of transformants” in the legend. Should the S15 row below the graph show “-+ -+ -+ -+”? So, one experiment without and one with S15 for each TniQ variant?

We have corrected this error. We also added a description of the transposition assay in the Figure 2 legend:

“In vitro transposition activity was monitored by transforming the reaction product into competent cells, and selecting the transformants on the antibiotics-containing plate (See Methods for detail). The number of colonies is plotted for each condition tested. Data are represented by the mean; error bars indicate SD (n = 3, biological triplicates). Raw data points are shown in red.”

- Page 6, line 144: In addition to the authors’ own publication, the TnsB STC structure papers (from the Montoya and Nowotny labs), which appeared at about the same time, should also be cited.

We have added the suggested references.

- Page 12, lines 298-300: I am not sure how the authors come to the conclusion about the role of TnsC R182 in filament disassembly. Please clarify.

We realize the focus on R182 was too restrictive, and it appears that additional residues are playing important roles in the transpososome. We have removed the specific description of the role of R182 residue and revised the section to accommodate the contributions from other residues. Below is the revised description (Page 12, Lines 306 – 310)

“We hypothesize that two turns of TnsC are required for transpososome formation with the ShCAST element (Fig. 6d), a state resistant to TnsB-mediated ATP hydrolysis (Fig. 6e) and stabilized by new TnsC-DNA contacts, K103, T121, K119, and R182. These contacts appear to stabilize a TnsC configuration that is not fully engaged with DNA compared to the helical TnsC formed outside the transpososome.”

Referee #3:

We already have a detailed view of Tn7 transposition. Meanwhile, the related CAST systems have been painted with broad brush strokes. This manuscript presents important new detail of a CAST system, which advances beyond isolated subunit structures and blob diagrams illustrating their interactions.

The data in Fig. 3 (and lines 174-200) provide insight into the TnsC–TnsC interaction. My only criticism is that the failure of a fragment to stimulate ATPase activity could have many explanations.

While we agree that there is more than one possible explanation for the ATPase assay, there is precedence for this idea in the literature. A short peptide from the C-terminal end of TnsB from prototypic Tn7 is sufficient to stimulate disassembly of TnsC (Skelding et al. EMBO J 2003). We have added a brief explanation of the provenance of this idea in the text (Page 7, Lines 177-178):

" In prototypic Tn7, C-terminal fragments are sufficient to stimulate TnsC ATPase activity²⁸."

The deviation of TnsC from a strict helical structure (lines 210–226) seems like a small point, and the significance was unclear to me, except in so far as it was different from a previous structure. The significance became clearer in the following section, and I wondered why the sections were separate.

We have combined the two sections as suggested.

Since I have a background in protein biochemistry, I tend to worry about the existence of unseen intermediates and the significance of structural snapshots. Notwithstanding, I do appreciate the insights provided by the present structure.

We agree that structural snapshots cannot capture the transient states that may be important to consider, however these transient states are exceedingly difficult to visualize and would constitute both a technical and scientific breakthrough. We appreciate reviewer 3's acknowledgment of the importance of the structural observations we report here.

The model in Fig. 6 is beautiful and represents a significant advance. However, the DNA helix, although stylized, is left-handed. The insights are developed and contextualized in an interesting discussion section.

We have fixed this figure so that the DNA helix is right-handed.

In the transposition literature the word 'transpososome' generally indicates a complex between the transposase and two transposon ends. Here it is defined in the abstract as an integration complex, which is more restrictive than how it is generally understood.

To our best knowledge, the transpososome refers to all the components required for integration. We have changed reference of integration complex to holo integration complex to better represent the idea that all required components for integration are present in the holo integration complex, i.e. transpososome.

Line 17 and elsewhere: THE AAA+ regulator: it is distracting if you drop the definite article.

We appreciate reviewer 3's attention to detail and have revised the main text for clarity.

Lines 17-18: Orientation with respect to what? What is meant by the length of TnsC? Is it a length in Angstroms? Polarity with respect to what?

We now appreciate the confusion brought about by the use of two different but similar terms (orientation vs polarity). In this context orientation refers to the dedicated protein-protein interaction interfaces of TnsC located at the N- and C-terminal faces. Polarity refers to the direction of the TnsC filament. Length here refers to the two turns of TnsC observed in the transpososome. With this feedback we realize the confusing nature of the abstract as originally written and have revised it significantly to clarify these concepts.

We have also replaced the term polarity with the more specific term direction throughout, and removed reference to length instead referring to the specific number of TnsC subunits contributing to the oligomeric assemblies we observe.

Line 19, "THE transposase...": Again, the missing word makes me unsure of the meaning, and I have to read the sentence a couple of times more to make sure I have not misunderstood. In this particular sentence, the problem is compounded by omitting THAT from the phrase "interactions we observe". Already, I'm finding this manuscript hard going. How can you know that the observed EM interactions stimulate the ATPase activity? Could it not be owing to an unseen intermediate?

Because TnsB and TnsC stably interact in the transpososome, we know the specific interactions we observe are not responsible for stimulating ATPase activity, because this structure represents a state of TnsC that is incapable of hydrolyzing ATP. Similar questions were brought up by reviewer 2 (see points 4 and 5 from reviewer 2). However, we also know that ATPase activity must have occurred at some point to result in the formation of the transpososome, because we observe a heterogeneous assembly of TnsC filaments prior to the addition of TnsB (described in response to point 5 from reviewer 2). Reviewer 3 is correct that the interactions to stimulate ATPase activity are not captured here. We have hypothesized that proximity of TnsB domain II β to the ATP-binding pocket of TnsC suggests that this domain might also contribute to ATPase activation. This is supported by reduction of ATPase stimulatory activity of TnsB mutants, however more work is required to determine the exact mechanism of ATPase activation. We hope the revised abstract and main text makes these concepts clearer.

Line 21: I could understand how a protein might 'track' along DNA, but I'm at a loss to know the meaning of "DNA does not track WITH TnsC protomers." From the context, I suspect that the authors are trying to say that a previously characterized TnsC homolog tracked round the DNA helix but that the new structure is different.

Yes, that is correct. We apologize for the lack of clarity in the abstract and have revised this in accordance with reviewer feedback.

Line 37: An alternative definition of the 'transpososome' to that offered in Line 14.

We have revised terminology throughout the text. The term transpososome should refer to all components required for integration at the target site.

Lines 38, 42: More missing definite articles. I won't mention this again.

Again we apologize for the lack of clarity and have revised the text to be clearer.

Line 43: "in some cases WAS shown to require". An experimental result should be stated in the past tense. Was shown because the act of showing was done in the past.

See above.

Line 55, 'obtained a high resolution structure': Was it not stated previously that there was more than one structural conformation resolved? It is best not to mention the minor structure until the point in the results where it is addressed, i.e. line 93.

We have moved mention of the minor structure to the end of the results where it is discussed in detail. See section titled: "Basis of insertion spacing variability", starting on Page 11 Lines 277.

Line 164, "TnsC's": Here and elsewhere, it is best to avoid using the possessive form of an inanimate object. Just write "stimulate the ATPase activity of TnsC". This even saves you five characters!

We appreciate reviewer 3's advice and have implemented these changes.

Lines 175-181: This section contains six repetitions of "ATP hydrolysis activity." You'll save a lot of characters by replacing it by ATPase and even more by eliminating some of the repetition. At about this point I began to become irritated by the use of the term "construct", which is lab slang.

All of these critiques relate to the lack of clarity in the text, so we have rewritten the text for clarity of language and incorporated this specific suggestion.

Line 32: In the transposition field 'domestication' is generally used to mean that a component has become a bona fide member of the community of host genes, where it performs a function and remains under purifying selection.

We thank the reviewer for pointing this out. We have revised this term to "acquisition" instead of "domestication" (Page 2, Line 36)

Line 39: What is the meaning of the slash in TniQ/TnsD? If it was tniQ-TnsD, I think most readers would take it as a heterodimer.

We agree with the reviewer that this can be confusing to readers. TnsD and TniQ are homologs, which we have explained in the main text instead of indicating with a slash.

Line 43: "in some cases WAS shown to require". An experimental result should be stated in the past tense. Was shown because the act of showing was done in the past.

We have revised this section for clarity, and this sentence was removed.

Line 44: The quotes round 'matchmaker' are an admission that it is slang. Consider using "bridge."

TnsC is commonly referred to as a molecular matchmaker in the field (without quotes) as described in other publications (Shen et al., *NSMB*, 2022). We have removed the quotes.

Line 58: The fact that it is orientation-specific seems hardly worth mentioning. This was established in the 2019 CAST in vitro reconstitution. Even the dCas9-mariner and dCas9-casposon papers from 2019 and 2021 were orientation-specific. It could hardly be otherwise.

Orientation in this context refers to the association of subunits. To the best of our understanding, genetic fusions necessarily restrict orientation of components, which isn't present in this system. In Tn7 it has been hypothesized but never shown that there are dedicated protein-protein interaction interfaces, which is shown for the first time here. We also realize the confusion brought about by using the term orientation in two different contexts (one referring to insertion orientation and the other referring to architectural features) and have clarified orientation to refer only to orientation of transposon insertions.

First paragraph of the results: if you have space consider citing, or at least mentioning, the development of the integration complex mimic approach. It might be worth devoting a sentence or two, here or in the discussion, about the integration-complex mimic-approach used here. Transpososomes tend to become increasingly stable as the reaction progresses through the intermediate stages. The approach goes back to the disintegration assays for HIV and V(D)J recombination (see 10.1093/emboj/cdf425 for references and discussion of conformational dynamics).

We appreciate reviewer 3's comment here, which reflects a deep understanding of the mechanistic insights we employed to reconstitute the transpososome. We have added the citation of the studies (Maertens et al., *Nature*, 2010; Yin and Craigie, *Protein Science*, 2012) that used the DNA design that mimics strand-transfer product as follows (Page 4, Lines 78 – 81):

“Because transposition is driven primarily via protein-DNA interactions, the transpososome is the most stable structure in the transposition pathway. This principle was successfully employed in the past to stabilize strand-transfer complex structures of other integrases/transposases, such as PFV^{20,21}.“

Line 107: TnsC DOES not FORM. This is followed by another missing word that makes the sentence distracting.

We have revised the wording to clarify our ideas (Page 5, Line 116):

“TnsC protomers lacking productive interactions with target site proteins are presumably disassembled by TnsB”

Line 111: Change to “PROBABLY stabilized”.

See our response to reviewer 2, point 5. Because the target pot contains a heterogeneous distribution of TnsC filaments, we know TnsB-promoted disassembly of TnsC filaments has happened to make the ensemble of transpososome particles homogeneous. However, the final two turns of the TnsC oligomer resist further disassembly because of the stabilizing interactions with target site-associated proteins. Therefore, we believe this point can be made without qualifiers. However, we agree with the reviewer that there are networks of interactions that stabilize the TnsC oligomer, in addition to the protein-protein interaction with the target-site-associated proteins. We have revised the sentence to include other possible interactions (Page 5, Lines 119 – 121):

“Therefore, the interactions between TnsC and target site associated proteins (Cas12k, TniQ, and S15) are stabilized against further disassembly by TnsB which is likely due to interactions at the target site(Fig. 1a, right). ”

Also, consider speculating on the mechanism of stabilization. Does the structure suggest a mechanism? Be careful to avoid tautological reasoning.

We could speculate on many possible mechanisms based on what we see in transpososome structure, which include new protein-DNA interactions, protein-protein interactions, DNA mechanics, and/or subtle allosteric effects.

We are very excited about these possibilities, which will require further investigation in order to comment on definitively. Unfortunately, due to length restrictions we cannot devote additional discussion to exploring these possibilities in the main text.

Line 128: The word ‘interaction’ is repeated three times in this sentence.

We have revised the sentence to address reviewer’s comment (Page 6, Lines 132 – 134)

“TniQ bridges the two TnsC protomers closest to Cas12k (Fig. 2a & Supplementary Fig. 2), however the TniQ-TnsC12 interface is much smaller than the TniQ-TnsC11 interface (325 Å² vs 915 Å², see Methods).”

Line 130, “reported importance”: Is there some uncertainty here? Do you mean ‘putative importance’?

We agree with the reviewer that the word reported might be confusing readers. We have removed this particular word.

Line 131: Either add supplemental data regarding S15 activity or state it as “data not shown”.

Our *in vitro* transposition data (Fig. 2d) shows that the addition of S15 improves transposition activity in all the conditions except for the negative control (when we omitted transposase). We have added the callout to figure 2d in the text (Page 6, Line 143).

Line 143, “RNaseH transposase”: Will the general reader understand that it has an RNaseH-like fold?

We agree that it is best to be more specific and have changed the text to instead refer to TnsB as a DDE/D transposase as follows (Page 7, Line 154):

“TnsB belongs to the family of DDE/D transposase, and bears significant similarities to MuA from bacteriophage Mu.”

Line 145: Is it necessary to always refer to TnsC as “the AAA+ regulation”? It seems repetitious and the significance of the emphasis is unclear.

We agree. The role of TnsC has already been introduced at this point. We have removed the “AAA+ regulator” in the sentence.

Line 149: How predicted? One would expect a flexible linker to be disordered because it would not otherwise be flexible. Sorry, I’m just saying that it’s hardly worth mentioning.

These residues were not only missing in the previous TnsB structure (Park et al. *PNAS* 2022) but also predicted to be disordered based on the primary sequence (as shown in Supplementary Figure 3). However, 25 residues of these disordered residues form structured interactions at the TnsB-TnsC interface in the transpososome, which we find a significant point.

We agree with reviewer 3 that this point was not clear enough. We have revised the section to clearly explain this rationale (Page 7, Lines 159 – 169).

Lines 149–152: The hook seems to be defined more than once.

We thank the reviewer for pointing out this mistake. We have removed the duplicated definitions of TnsB^{Hook}.

Line 167: Excitingly? Significantly might be better. Less is more.

We have changed the word to "notably", in line with the suggestion.

Line 168, "we observe": I think "there is" is more impactful.

We have made this change.

Line 172: Another set of quotes acknowledging the use of slang. Transduce?

We have changed the word "sense" to "recognize" and removed the quotes.

Line 179, "mutant TnsB insertions": Is mutant TnsB being inserted into something?

We agree with the reviewer that the sentence was confusing. We have revised it to following (Page 9, Line 208):

"It is particularly striking that, in the absence of S15, WT TnsB retains high levels of on-site targeting (69%) whereas TnsB mutants have no targeting ability (Extended Data Fig. 8).

Line 245: Citation needed for "previously identified DNA binding residues.

We have added the proper citation as suggested (Page 10, Line 251).